# Selection of Promising Novel Fragment Sized *S. aureus* SrtA Noncovalent Inhibitors Based on QSAR and Docking Modeling Studies

**DOI:** 10.3390/molecules26247677

**Published:** 2021-12-19

**Authors:** Dmitry A. Shulga, Konstantin V. Kudryavtsev

**Affiliations:** 1Department of Chemistry, Lomonosov Moscow State University, Leninskie Gory 1/3, 119991 Moscow, Russia; 2Laboratory of Molecular Pharmacology, Pirogov Russian National Research Medical University, Ostrovityanova Street 1, 117997 Moscow, Russia; 3Laboratory of Chemical Synthesis and Catalysis, Moscow Institute of Physics and Technology, Institutskiy per. 9, Moscow Region, 141701 Dolgoprudny, Russia

**Keywords:** sortase a inhibitors, StrA inhibitors, *Staphylococcus* *aureus*, antivirulent agents, small-molecule drugs, fragment-based drug discovery, QSAR, descriptor selection, classification model, regression model, virtual screening, molecular docking, machine learning

## Abstract

Sortase A (SrtA) of *Staphylococcus aureus* has been identified as a promising target to a new type of antivirulent drugs, and therefore, the design of lead molecules with a low nanomolar range of activity and suitable drug-like properties is important. In this work, we aimed at identifying new fragment-sized starting points to design new noncovalent *S. aureus* SrtA inhibitors by making use of the dedicated molecular motif, 5-arylpyrrolidine-2-carboxylate, which has been previously shown to be significant for covalent binding SrtA inhibitors. To this end, an in silico approach combining QSAR and molecular docking studies was used. The known SrtA inhibitors from the ChEMBL database with diverse scaffolds were first employed to derive descriptors and interpret their significance and correlation to activity. Then, the classification and regression QSAR models were built, which were used for rough ranking of the virtual library of the synthetically feasible compounds containing the dedicated motif. Additionally, the virtual library compounds were docked into the “activated” model of SrtA (PDB:2KID). The consensus ranking of the virtual library resulted in the most promising structures, which will be subject to further synthesis and experimental testing in order to establish new fragment-like molecules for further development into antivirulent drugs.

## 1. Introduction

Antibiotics are widely used for medicinal indications to increase the success rate of medical treatment. However multidrug-resistant (MDR) strains of bacteria have become one of the more well-known problems in hospitals [1]. The main cause of MDR is the extensive use of antibiotics disrupting the key biochemical pathways of bacteria. The latter results in high evolution pressure forcing bacteria to quickly find a workaround via spurring the natural selection mechanism. Often, even a single nucleotide mutation is sufficient to obtain drug resistance for the wild type of a bacteria. A new approach should be used since the human ability—both in quality and speed—to produce new antibiotics is intrinsically inferior compared to that of bacteria. One of such promising approaches [2] is to reduce the evolution pressure of the drugs at the same time as reducing the exponential proliferation of bacteria in patients, causing negative effects. Thus, the new drugs should not try to completely kill the bacteria but, rather, to diminish their virulence. In order to exploit this idea in practice, a promising drug target should be identified that could be used to design new antivirulent drugs [3].

*S. aureus* is one of the major known contributors to the MDR bacteria family. Sortase factor A (StrA) of *S. aureus* is generally validated as a promising target for antivirulent drugs [4,5,6,7,8]. It was shown, in particular, that a selective irreversible inhibition of SrtA leads to diminished bacteria proliferation. Another evidence is the experimentally observed diminished virulence for StrA knockout *S. aureus* strains [9].

SrtA is a cysteine transpeptidase, which provides a mechanism for a bacteria cell to display different surface proteins on its outer surface. Many of these extracellular proteins are involved in virulence functions, such as evasion of the immune system and better adhesion to the host cells. SrtA acts using a two-stage ping-pong mechanism [10,11]. At the first stage, the active site Cys-184 specifically cleaves a peptide bond between the Thr (T) and Gly (G) residues of a general LPXTG (where X is any amino acid) amino acid “sorting” sequence, with the Cys-184 forming a thioester acyl enzyme intermediate with a Thr residue of a surface protein held exposed to the bacteria cell surface. At the second stage, this intermediate is resolved by an attack of the amino group of pentaglycine-branched lipid II, essential for cell wall peptidoglycan synthesis.

The similar class of cysteine proteases is absent in humans and eukaryotes in general [12], thus theoretically opening the way for developing more selective and less toxic drugs. Another advantage is that StrA represents an outer surface-bound protein, which is far more accessible to drugs compared to inner bacteria cell targets. The ongoing difficulty associated with this target is the significant mobility of the certain enzyme domains, which form the binding site [13,14,15]. In particular, state-of-the-art molecular dynamics simulation studies [14,15] confirmed the large-scale motions of β6/β7 and β7/β8 loops with conformation transitions taking place in a micro-to-millisecond timescale found by NMR earlier [13] and revealed the multiple binding conformations available for sorting the LPXTG signal. Perhaps that is the main reason why, despite the work still being in progress, no molecule has been identified as active through non-covalent interactions in a nM concentration. Recently, several classes of both covalent and noncovalent active against SrtA compounds have been identified [4,5,7,16,17,18,19].

The main objective of this work is to use the earlier identified known hits from the literature [4], as well as the combined ligand-based and structure-based approach, to identify new fragment-sized entities for further rational development. It will potentially open new opportunities to design new antivirulent antibiotics using both fragment-based drug discovery (FBDD) [20,21,22] as well as the more traditional drug discovery tools.

We use a series of compounds containing the dedicated motif, 5-arylpyrrolidine-2-carboxylate, which has been previously shown to be significant for covalently binding SrtA inhibitors [23] as potential ligands. It is expected that the ability of the covalently binding SrtA inhibitors containing the dedicated motif to recognize that the SrtA target could be useful to stem from them the development of noncovalent inhibitors.

In what follows, firstly, the known molecules from the ChEMBL database [24] with *S. aureus* SrtA activity are used to build both classification and regression quantitative structure–activity relationship (QSAR) models. Alongside this, interpretation of the descriptor correlations with activity and each other is also provided. Secondly, the QSAR models built are utilized to make predictions for SrtA inhibitory activity for the prospective ligand list with the dedicated motif. Thirdly, the smallest known actives from ChEMBL, filtered out as obeying the rule of three (Ro3) [25,26], are docked to the activated form of the SrtA target, followed by the analysis of the binding modes. After that, the molecules from the prospective ligand list are also docked to the target using both of the available enantiomer geometries. Finally, a consensus selection of the ligands most promising for further study is made using the classification and regression QSAR model predictions, docking scores and estimated ligand efficiencies, as well as other calculated properties contributing to a molecule to be drug-like—the components of the rule of five [27,28] (Ro5) filter.

## 2. Materials and Methods

### 2.1. Virtual Drug Ligand List

Based on the previous experience [29,30,31,32,33,34,35,36,37] and the available synthetic schemes, a set of potential ligands has been composed (Table 1) in order to select the most promising ones by virtual screening and docking for synthesis and antiadhesive property studies.

### 2.2. Virtual Screening

The available data for ligands active against *S. aureus* StrA were taken from ChEMBL [24]. In this work, we decided to take structures with activity in a high concentration range as “true inactives” instead of using random structures, as it is often done in cases when inactive structures are absent or scarce. Both the classification and regression models were built to potentially increase the robustness of the predictions using consensus scoring. It should be noted here that the models built are not expected to be solely reliable and selective models; rather, they are intended to be used as a focused filter based on general molecular properties. One of the main reasons, the chemical structures of the known actives are highly heterogeneous and not numerous, so the adequate coverage of the chemical space is questionable. Another reason, which is related to the previous one, is that the binding site, at least in the open form of the enzyme [14], is rather extended, leaving room for multiple plausible binding positions within it for different small molecules [19]. Thus, a proper predictive QSAR model, based on detailed structure descriptors, is hardly warranted. Nonetheless, a model built on mostly physicochemical descriptors is likely to capture the generalized requirements of the binding site and could be subsequently used to narrow down the chemical space by filtering out structures that differ significantly in their properties from the known actives. Additionally, the relative importance of the descriptor, as well as the sign of the coefficient, are also instructive to build meaningful interpretation of the models.

In building correlation models, a special emphasis was put on the robustness of the models, which should be more generalizing than quantitatively accurate on very limited and highly skewed molecule sets. This was done via the following experiment design decisions. Firstly, due to scarcity and high scaffold heterogeneity and bias of the molecule set, no split into training and test sets was done. Secondly, among the types of regressions and classification techniques, the choice was made in favor of not excessively flexible and adaptable models, such as different types of neural networks or gradient boosting tree-based models, which have found numerous applications but require much care in order not to overfit the model. In particular, to properly train the above models, larger and much more representative datasets would be required. Finally, the variants of the algorithms with an option of regularization, i.e., penalization of absolute values of the coefficient descriptors, were used to build the models. An additional benefit of using the regularized versions is their ability to naturally conduct selection of the significant descriptors, leaving insignificant ones outside the model.

In its extreme, the models should be as simple and broadly applicable as widely used Lipinski’s “rule of five” (Ro5) [27,28], “rule of three” (Ro3) [25,26] and others [38,39].

All the models were built by means of the Scikit Learn [40] and NumPy [41] libraries. Three-dimensional molecule visualization was done using PyMol [42].

#### 2.2.1. Descriptors

For our study, molecular descriptors should roughly distinguish molecules by their properties. A set of descriptors used in the study (Table 2) reflected the basic composition, physicochemical properties and size of the molecules. On the other hand, a set of topological descriptors were also added into the descriptor set, since, despite their simplicity, those descriptors have been previously capable of describing difficult-to-model properties still dependent on the molecule structure. All the descriptor values were obtained using RDKit [43].

Prior to use, the descriptor values were normalized to the zero mean and unit standard deviation based on the molecule set used to build the model.

Within the work, a correlation of the descriptor values with the modeled activity was analyzed. After that, the descriptors were used to produce QSAR models. The final working set of descriptors for each model was defined based on mutual descriptor correlations (only descriptors having mutual correlations of less than 0.7 were left) and the use of the regularization technique, which helped to eliminate less significant descriptors to make models more robust.

#### 2.2.2. Classification Models

A dataset of structures for building classification models was extracted by two separate queries from the ChEMBL database to obtain active (Appendix A) and (true) inactive (Appendix A) structure lists. To extract the structures related to activity against the StrA of *S. aureus* target_chembl_id = CHEMBL5362 was used. An active set was defined as structures having MW < 1000, activity type of the assay inhibition (standard_type = “IC50” or “Inhibition” or “Ki” or “Km”) and the well-defined activity (standard_relation = “=”). The resulting standard_units value was either “nM” or “%” or “ug.mL−1”. After the removal of duplicates and mercury-containing species nonselectively binding to the catalytic cysteine (Cys-184), the active set included 118 structures (Appendix A). 

A nonactive list was composed with similar criteria but using standard_relation = ”>”, which means the structures did not reveal any activity at the concentration used for screening. The resulting structure had standard_type = ”IC50” and standard_units in either “nM” or “ug.mL^−1^”. After duplicate removal, the inactive set contained 72 structures (Appendix A). For both sets, charged forms of the structures were cast as neutral ones with removal of the specified counter ion.

Two methods were used to build classification models: support vector machine (SVM) [44] and logistic regression (LR) in their Scikit Learn [40] implementations sklearn.svm.LinearSVC and sklearn.linear_model.LogisticRegression, respectively. In order to eliminate nonsignificant descriptors, L1 regularization was used for all methods.

The area under the curve (AUC) for the receiver operator characteristic (ROC) as well as the number of true positives (TP), true negatives (TN), false positives (FP) and false negatives (FN) are used as the usual metrics for the classification models.

#### 2.2.3. Regression Models

A dataset of the structures for building regression models was also extracted from the ChEMBL database with the following query: MW < 1000 Da, activity = “IC50” and activity units “nM” with the additional criterion of standard_relation “=”, retrieving only structures with well-defined activity. After the removal of mercury-containing molecules, applying geometric mean averaging for duplicated data and the transformation of charged forms into neutral, a set of 86 structures was obtained (Appendix A).

Since the list of structures with numerical activity is rather scarce and their activity range is narrow (no highly active compounds are known, and only compounds with the above threshold of the assay activity are presented), we decided to enrich the molecule set with structures that had not demonstrated activity to SrtA in the range of the concentrations studied by assays. To that end, an additional set of structures was extracted from ChEMBL with the following query: activity = “IC50”, activity units is “nM” and “standard_relation” is “>”. After manual filtration, a set of 45 structures was obtained (Appendix A). Since these values are basically the higher threshold concentration detected by the assays used, the true activity of those structures could span the range from just above the specified threshold values to milli- and molar values (practically not the inhibitor). It is highly unlikely that the true activities were close to the array concentration threshold values. Therefore, a conservative estimate of the numerical value of activity was made by multiplying by 10 the activity data extracted from ChEMBL (Appendix A). It should be noted that a higher value of the multiplier would effectively render the task as a classification, since the activity range would differ significantly for active and inactive structures. Thus, a set of 131 structures was composed for the regression analysis study.

Activity units for all the structures were converted to an energy-like log10 scale, which is a common practice for building QSAR models.

A preliminary choice of the descriptor set was done by using the Recursive feature elimination method (sklearn.feature_selection.RFE) in combination with LASSO regression (sklearn.linear_model.Lasso) as a model.

Two types of models were built to predict the activity for the prospective set of structures: ‘‘least absolute shrinkage and selection operator” (LASSO) regression [45], which is a variant of classical regression with L1 regularization applied to the parameter values, and a modification of support vector machine (SVM) for regression (SVR) [46] with a nonlinear kernel (sklearn.svm.SVR(kernel = ”rbf”)). The latter model should be prone to overfitting for the structure and descriptor sets used but was retained for the larger diversification of the predictive models in order to make a consensus prediction more robust.

The root mean squared error (RMSE) of prediction as well as the coefficient of determination of linear regression, *R*^2^, are used as usual metrics for the regression models. Additionally, the leave-one-out (LOO) cross validation counterparts, *R*^2^_LOO_ and RMSE_LOO_, were used to roughly estimate the predictive power of the models (details in Section 2.2.4).

#### 2.2.4. Statistical Estimation of Models

In addition to the statistical performance of the molecules sets described, three types of additional statistical analyses were done for the final models. Firstly, leave-one-out (LOO) statistics was obtained for the best of both the classification and regression models to estimate the signs of potential “overfitting”. For this reason, N models (where N is the number of molecules in the molecule set) were independently built using N-1 molecules to predict a value of the molecule not included in the training set. 

Secondly, the bootstrap method [47] was used to estimate the confidence limits of support vector machine (SVM), logistic regression (LR) and least absolute shrinkage and selection operator (LASSO). It was not used for support vector machine for regression (SVR), since this method does not provide its coefficient for straightforward interpretation. In the bootstrap method, we built 1000 models based on synthetically generated datasets consisting of random choices of the molecules from the training set. The confidence level of each parameter value for the set of 1000 parameter sets obtained was estimated at the 95% level, i.e., the parameter values were provided for the 2.5 and 97.5 percentiles of the parameter distributions. All calculations were done using Scikit Learn [40] and NumPy [41].

Finally, a brief scan for the regularization parameter value was done for all the models (Appendix A) with the aim of making a sensible choice in favor of a more generalizing model instead of the most accurate one for a particular structure set, which we believe is highly biased. In these scans, the statistical performance obtained in the training set is compared to the performance estimated by leave-one-out (LOO) metrics as an affordable measure of the predicting power in a scarce dataset of the structures at hand.

### 2.3. Docking Studies

#### 2.3.1. Receptor Model

The “activated” form of *S. aureus* SrtA_ΔN59_ (PDB:2KID) was chosen for two reasons among the available experimental structures in the Protein Data Bank (PDB) [48]. Firstly, it provides the most compact site, which better fits the small fragment-sized molecules. Secondly, this structure was successfully employed to rationalize the activity of compounds in several studies [49,50,51,52], notably a recent work revealing Tideglusib and its analogs [16] as potent to SrtA and not generally toxic to bacteria.

The first of the 20 available NMR modes was used. The receptor was stripped off the ligand by breaking the disulfide bond with Cys184 and then prepared for docking using MGLTools 1.5.6 [53] with the standard settings.

#### 2.3.2. Docking Protocol

Docking was done by means of AutoDock Vina v1.1.2 [54]. All the studied ligands were prepared with help from MGLTools 1.5.6 [53] using the standard settings. The 3D geometries for all the ligands were generated using --gen3D option of OpenBabel v3.0.0 [55].

The docking protocol was slightly modified compared to the standard one by increasing the exhaustiveness value from 8 to 32 and increasing the number of output binding modes from 9 to 15 to better explore the ligand–receptor configuration space for fragment-sized molecules.

#### 2.3.3. Docking of the Ro3 Subset of ChEMBL Ligands

Prior to docking the ligands from the prospective ligand list, the possible preferred modes of interactions between the known active compounds and SrtA were studied. To this end, a subset of fragment-like compounds complying to the “Rule of 3” (Ro3) [25,26] was selected from the active compounds used for classification model building (Figure 1). The Ro3 are defined as MW < 300, HBD ≤ 3, HBA ≤ 3 and cLogP ≤ 3.0. All the descriptor calculations and molecule filtering were done using RDKit [43]. The resulting 10 compounds were docked according to the protocol described, and then, the obtained binding modes were analyzed.

## 3. Results and Discussion

### 3.1. Virtual Screening

#### 3.1.1. Classification Models

##### Descriptors

The full set of descriptors was filtered out in order to discard both those poorly correlated to the activity descriptors, as well as the highly correlated pairs of descriptors.

To begin with, the correlation of the descriptors with the activity (values: 0—for inactive and 1—for active structures of the set) was studied (Appendix A) in order to find the most correlated descriptors and make up an initial interpretation for those correlations. 

Reasonable interpretations were obtained for the influence of the estimated physicochemical properties of a molecule on its predicted activity. For instance, an increase in lipophilicity (MolLogP) above the mean value of 3.33 adversary affects activity, as do the properties closely related to the general size of a molecule: NumAtoms, MolWt and MolMR when increased based on their mean values of 45.0, 357.2 and 96.0, respectively. Therefore, a large and lipophilic molecule is not beneficial for activity. To extend the above further, the increase of the polar surface (TPSA) and the number of heteroatoms (NumHeteroatoms) leads to higher chances for a molecule to be active for the molecule set studied. Interestingly, there is an asymmetry regarding the influence of changing the number of hydrogen bond donors and acceptors. Thus, an increase of the number of hydrogen bond donors, NHOHCount and NumHDonors, above the mean numbers of 1.8 and 1.8 lowers the chances of a molecule to be active. Meanwhile, an increase of the number of hydrogen bond acceptors, NOCount and NumHAcceptors, above the mean values of 5.0 and 4.3 beneficially affects the probability of a molecule being active. Consequently, an active molecule should contain a significant polar surface represented mainly with acceptors of hydrogen bonds. This observation might be interpreted in terms of the structural requirements that arise due to the binding site constitution. It is well-established that Arg197 is critical to the binding and functioning of SrtA, and it has been also determined that it acts more like a hydrogen bond donor than as a charged species [56]. It is also corroborated by the not necessarily charged nature of the natural LPXTG-containing ligands. We therefore hypothesize that, in order to coordinate the charged and hydrogen donor Arg197 residue, a ligand should be more like a dipolar aprotic solvent, i.e., it should contain more hydrogen bond acceptors and as few as possible hydrogen bond donors clearly placed in certain positions. To further support this hypothesis, descriptors that reflect the articulated partial atomic charge, MaxPartialCharge, MinPartialCharge and MaxAbsPartialCharge, negatively correlate with the predicted activity.

Another observation is an increase of the number of aromatic rings, NumAromaticRings, and the number of chiral centers, NumChiralCenters, above the mean values of 2.0 and 1.3 is not beneficial for activity. Interestingly, the number of aliphatic rings, NumAliphaticRings, appears to be not significant to determine activity. Excessive conformational flexibility with NumRotatableBonds above the mean value of 5.4 is also not favored for activity.

It is instructive to note that the mean values of descriptors in the molecules set are well within the requirements for Lipinski’s “Ro5” [27,28], Ghose [38] and Veber [39] filters for druglike molecules. However, as we are interested in fragment-sized lead-like compounds, the “rule of three” (Ro3) [25,26] is a more appropriate filter. Its requirements are stretched as judged by the mean descriptor values for the set (Appendix A).

The next step was to exclude the mutually correlated descriptors in order to build more reliable predictive models. To that end, the correlation between the descriptor values on the molecule set was studied. Firstly, a strong correlation was observed between TPSA, NHOHCount, NOCount and NumHeteroatoms. A significant correlation was present between NHOHCount and Kappa1. Another rather predictable correlation was between NumHAcceptors and TPSA, NOCount and NumHeteroatoms. The same was for the NumHDonors and NHOHCount pair. In order to maximize the interpretability, a combination of the descriptors, capturing the general number of heteroatoms, such as TPSA (or, alternatively, NumHeteroatoms), and the hydrogen bond donor/acceptor differentiating descriptors, such as NumHDonors and NumHAcceptors, was decided to be used for further research.

Secondly, the topological indices studied in the work generally established good correlations with each other and with different descriptors. On the one hand, it once again confirmed the usefulness and efficacy of the topological index concept, which made it possible to establish decent quantitative correlations with the properties hardly expressed in terms of the simplest structural descriptors. On the other hand, in cases where the topological descriptors correlated with the more readily interpretable physicochemical or structural descriptors, the latter should be preferred for building the quantitative model to facilitate interpretation of the model. Perhaps a single exception is the Balaban topological connectivity index (BalabanJ), which does not correlate to other indices; however, its correlation to activity is also negligible. As for descriptors of the group Chi, they produce appreciable correlations with the descriptors depicting the general molecule size, such as NumAtoms, MolWt and MolMR. The same holds for Kappa* and BertzCT descriptors. In addition, Chi0 and HallKierAlpha also reasonably correlate to FractionCSP3. This analysis suggests that topological indices should be excluded from the descriptor set to build quantitative models for not bringing additional information in favor of more readily interpretable counterparts.

Finally, descriptors reflecting the general molecule size, such as NumAtoms, MolWt and MolMR, predictably form mutual correlations, so only a single representative, NumAtoms, was retained for the final descriptor set.

Based on the above analysis, 12 descriptors were chosen to be used in building quantitative models (Appendix A). It should be noted that the ratio of the number of endpoint data and the number of descriptors, 12–190 (118 actives and 78 inactives), reasonably corresponds to good practices of QSAR modeling [57].

##### Classification Models on Training Set

Two classification methods were studied in the work for the reasons described in the Materials and Methods: support vector machine (SVM) and logistic regression (LR), both with the L1 regularization of the model coefficients. The main benefit of L1 regularization is that it enables automatic feature (descriptor) selection, based on the significance for the model. At each regularization parameter value, only the coefficients at the most significant descriptors receive nonzero values. The larger the L1 penalty, the fewer descriptors enter the model. This results in a rougher model that is less accurate for the training data but, at the same time, is less prone to be overfitted.

We built three different models for each of the classification methods used, SVM or LR. The first one is a model with high regularization constant, containing only one descriptor, which reveals the most significant descriptor on the set and sets the lower limit to the model accuracy. The second one is built with a mild regularization constant, so that all descriptors that enter the model contribute to the prediction, which sets the upper limit in the accuracy as attainable for the specific molecule and descriptor sets. The third model, built with intermediate values of the regularization constant, is an optimal model containing only a subset of the initial descriptor set and combining reasonable accuracy and robustness.

Using the SVM method, a one-descriptor model contained only the MolLogP descriptor was the most discriminative among the others in the molecule set. Similar to the correlation analysis above, it enters in the model with a negative sign, meaning, generally, the more lipophilicity, the less likely a molecule is active. It is an unusual behavior that reflects the peculiarities of the active/inactive dataset molecules. The visual analysis of the MolLogP distribution among the active and inactive subsets supports the simple distinction (Figure 2) relative to the mean MolLogP value of 3.34, although the dispersion in each class is also appreciable. The area under the curve (AUC) for the receiver operator characteristic (ROC) for the one-descriptor SVM model is appreciable at 0.71 (grey in Figure 3), which also supports the discriminating power of the MolLogP descriptor for the molecule set studied. The confusion matrix parameters for this model are TP = 78, TN = 49, FN = 40 and FP = 23. 

A mildly regularized model with all 12 descriptors (chosen at the stage of the initial correlation analysis) results in a ROC AUC value of 0.83 (Figure 3 green) and sets the accuracy limit for the combination of the molecule set and SVM model type. This model is significantly better for early enrichment (left bottom corner of the ROC curve in Figure 3) compared to the one-descriptor model, despite the AUC increase not being dramatic. The confusion matrix parameters for this model are TP = 106, TN = 43, FN = 12 and FP = 29. 

The third model, built with intermediate values for L1 regularization of the parameters, uses effectively only nine out of 12 descriptors. It combines the AUC value, 0.81, close to the full 12-descriptor model, good early enrichment properties and, potentially, the higher robustness (more generalization) via the exclusion of less significant descriptors (Appendix A). The confusion matrix parameters for this model are TP = 103, TN = 40, FN = 15 and FP = 32. The signs of the coefficients at the descriptors for the SVM model and, hence, the interpretation of their influence are similar to those obtained at the study of the correlation of the descriptor value with binary activity. The leave-one-out (LOO) estimation of the AUC for this model is 0.76, close to the value for the whole training set. Finally, this nine-descriptor model was used to predict the activity for the molecules of the prospective ligand list.

Logistic regression (LR) was chosen as the second type of classification model. The setup, like the SVM model training experiment, was used to obtain three models with different levels of the regularization factor and the final number of descriptors used in a model. The results for the LR models were very close to the SVM models. Thus, the one-descriptor model contained only the MolLogP descriptor. The same three descriptors were excluded at the intermediate regularization constant value (Appendix A). All the signs of the coefficients at the descriptors were also the same. Additionally, the statistical performance was the same (Figure 3, right). The confusion matrix parameters for this model are TP = 78, TN = 49, FN = 40 and FP = 23. Similarly, the nine-descriptor model was used for further predictions on the prospective molecule list. The confusion matrix parameters for this model are TP = 103, TN = 39, FN = 15 and FP = 33. The confusion matrices for the 12-descriptor models for SVM and LR methods coincided. The LOO estimation of the AUC for the nine-descriptor modes is 0.75. The overall similarity in the quality of the prediction in the training set should be attributed not to the model peculiarities but, rather, to the quality of the input data. Therefore, the more flexible models would result in overfit and not in the accuracy gain. Thus, we expect a decent generalization was achieved by the finally chosen models, which lays the foundations for robust activity predictions for dissimilar molecules of the prospective molecule list.

##### Predictions for the Prospective Ligand List

The nine-descriptor SVM and LR classification models trained from the literature data were used to predict the activity of the structures from the prospective ligand list. Since the models do not distinguish enantiomers by design, all the following data were provided for the general composition formula of the molecules. The predictions appeared almost identical for the two models (Appendix A and two columns named “Classification” in the final decision Table 8), and even the ranks of the activity predictions (based on the model predicted internal values) were quite similar.

Firstly, 17 or 18 out of 24 molecules were predicted as active for the SVM and LR models, respectively. Secondly, the most active molecules, judged by each model predicted internal values or activity ranks, were KUD718, KUD138, KUD649, KUD833, KUD1066, KUD1130, KUD834 and KUD1044 (Appendix A). Thirdly, six structures: KUD1050, KUD759, KUD1133, KUD1135, KUD990 and KUD1022 (in the order of decreasing probability of being inactive) were predicted to be inactive by both the SVM and LR models. KUD1008 was predicted inactive by SMV and active by the LR model (Appendix A).

#### 3.1.2. Regression Models

##### Descriptors

Initially, the same set of 12 descriptors used to build the classification models was used for deriving and interpreting the correlation of the descriptors with activity. All the descriptor values in a set were normalized to a zero mean and unit standard deviation.

The correlation of each of the 12 descriptors in a training set is shown in Table 3. An interpretation is they are very close to the case of a classification model dataset and values. Firstly, an increase of the general lipophilicity of a molecule (MolLogP) generally leads to an increase of log10(IC50), i.e., a decrease in activity. Secondly, a mean polarity, judged by MaxPartialCharge, MinPartialCharge and MaxAbsPartialCharge, correlates to decreased activity. Thirdly, the preference for the generally nonaromatic nature of a SrtA active ligand is also revealed by the signs of the correlation of FractionCSP3, NumAliphaticRings and NumAromaticRings. Fourthly, an increase in the number of chiral centers in a molecular (NumChiralCenters) adversary affects the predicted activity. Finally, an excessive conformational flexibility (the sign at NumRotatableBonds) is also not welcome.

The different correlation patterns from the classification dataset are, however, observed for the hydrogen bond descriptors. Thus, the combination of the signs and values for the correlation of NumHAcceptors, NumHDonors and NumHeteroatoms suggests that an increase in the number of hydrogen bond donors results in an activity increase. The number of aliphatic rings also correlates in an adversary way to the activity.

An additional study revealed that the normalized distribution of the number of chiral centers (NumChiralCenters) contains outliers well beyond the sigma values (Figure 4). These outliers correspond to two molecules (Figure 5) that were excluded from the training set as highly nontypical. Consequently, a molecule set of 129 molecules was further used for the study.

The first regression models built using the descriptor set found earlier for the classification models study showed, however, that the regression model performance was below the reasonable expectations. We decided to select the new set of descriptors more optimal for regression model building. For that reason, the recursive feature elimination (RFE) technique, as implemented in sklearn.feature_selection.RFE, was used along with LASSO regression (sklearn.linear_model.Lasso) as the “estimator” regression model to select the 12 most significant descriptors (Table 4), which appeared to be excluded from the QSAR models upon their gradual simplification. The main difference compared to the previous descriptor set was the presence of three topological indices, Kappa1, BertzCT and Chi3v, in it. Thus, in order to achieve better performance, a certain degree of interpretability was sacrificed. Notably, the NumChiralCenters and closely related for this set FractionCSP3 descriptors remained the most discriminative. The set of MolWt, Kappa1, BertzCT, NumAliphaticRings and MolLogP described the size and the form of a molecule. The remaining set of descriptors: NOCount, TPSA, NumHeteroatoms and NumHAcceptors described the polar characteristics of an active molecule, which was significantly biased, as was previously noted in the classification model analysis, toward the hydrogen bond acceptor’s presence, with the same interpretation applicable.

##### Regression Models in the Training Set

Two types of regression models were built within the study. The first one, which is the most straightforward and interpretable, is LASSO regression (sklearn.linear_model.Lasso) with L1 regularization in the parameter values. The second one is the support vector-based regression model with a nonlinear radial basis function kernel (sklearn.svm.SVR(kernel = ”rbf”)). The latter model is expected to be less generalizing for the predictions in exchange for better accuracy for the molecules, similar for the molecules in the training set.

For LASSO regression, three models were built with different values of the regularization parameters, L1, similar to the case of the classification model study. In the single descriptor model, only the NumChiralCenters descriptor remained, emphasizing its significance for quantitatively ranging the activity for the molecule set studied.

The mildly regularized model with all 12 descriptors remaining in it generally shows a modest performance in terms of the quantitative models. However, it is capable of roughly ranging the molecules by potential activity (Figure 6, left). The intermediate value for L1 was chosen such that nine out of 12 descriptors had the nonzero values in it. As in the classification case, its performance is only slightly worse (Figure 6, right) compared to the mildly regularized 12-descriptor model, but a better model generalization is expected in exchange. 

The coefficients in the nine-descriptor model admit reasonable interpretation (Table 5). The coefficients at the TPSA, NOCount and NumHAcceptors descriptors show that the increase of the polar surface area due to an increase in the number of hydrogen bond acceptors leads to an increase in the predicted activity (lowering the logarithm of concentration). The values at the FractionCSP3, NumAliphaticRings and MolWt descriptors show that the growth of the molecule caused by the fragments with a moderate share of conjugated and aromatic atoms also results in activity gain, similar to the conclusions made using the classification model analysis. Finally, a simple increase of the molecule size, judged by MolWt, adversary affects the predicted activity. The model is not quantitatively accurate: *R*^2^ = 0.43, *R*^2^_LOO_ = 0.34, RMSE = 0.79 and RMSE_LOO_ = 0.84, but it was expected, and the aim was to build a model that was able to rank structures substantially different from the training set. Thus, the nine-descriptor model was further used for predictions for the prospective ligands.

For the second type of quantitative model, a support vector machine for a regression (SVR) model with a nonlinear kernel was chosen to enhance the diversity in the set of models, which predictions will be used in a consensus manner. A mild regularization factor was intentionally used (C = 20 for sklearn.svm.SVR) in order to obtain a model as accurate as possible in the training set still not wildly overfitted for the test set of molecules from the prospective ligand list. For this model, all 12 descriptors, selected during the RFE procedure, were used. As expected, the performance of this model in a training set of molecules was much better than for the linear models (Figure 7): *R*^2^ = 0.83, *R*^2^_LOO_ = 0.51, RMSE = 0.43 and RMSE_LOO_ = 0.74. Interestingly, the most significant outliers of the model were those for which the model was too optimistic in its predictions. This model was further used for the predictions of the prospective ligand list.

##### Predictions for the Prospective Ligand List

As for the classification models, the regression models were used to predict the most active and efficient ligands out of the prospective ligand list (Appendix A and the last four columns of the final decision Table 8). Along with the predicted activity, the ligand efficiency (LE) metric [58] was also provided to better emphasize the small-sized molecules, which predicted the activity was significant compared to their size. Those could be the high priority candidates for further rational development. As for the classification models, the data were provided for the molecule structures regardless of their chirality.

The two chosen regression models led to different predictions for activity in the prospective ligand list (Appendix A and the last four columns of the final decision Table 8). The SVR model was systematically more optimistic in its predictions, which was seen by comparing the distributions of the predicted activities made by the SVR and LASSO models (Figure 8 left). The latter also showed that the activity of the most active structures from the training set was higher than the predicted activities for the prospective ligand list, which represented a focused, accessible chemical space to study. On the other hand, the distributions of the LE values for the training set and both predictions in the test set were much closer to each other (Figure 8, right). It is also interesting to note that the correlation of the activities predicted by the two models was rather low—the Pearson correlation coefficient was 0.33, whereas the correlation between the predicted LE values was significant—0.91. This means that the models tended to produce different predictions mostly for large molecules, whereas, for small-sized molecules, the predictions were closer.

Strikingly, using the consensus of the two model estimations, the predicted most active molecules list coincides with the highest predicted LE molecule list, which is KUD138, KUD165, KUD224, KUD225, KUD718, KUD1130, KUD1132 and KUD1134 (Appendix A). 

The predicted least active and least ligand efficient (LE) molecules are also very close to each other (Appendix A). The least active molecules predicted are KUD834, KUD990, KUD1008, KUD1022, KUD1044 and KUD1050. The least LE molecules predicted are KUD759, KUD834, KUD990, KUD1008, KUD1022 and KUD1050. The interpretation is similar to the classification case: the excessively large and containing numerous chiral centers molecules are disfavored by the models. It should be noted, however, that the models were built using rather scarce available experimental data, which were likely to be significantly biased in terms of the chemical space. Thus, the predictions, which are, by design, close to the simple physicochemical property filters, should be used for the recommendation basis unless more definite predictions are available.

### 3.2. Docking

#### 3.2.1. Ro3 Subset of ChEMBL Ligands

The docking study of the Ro3 compatible subset of the ChEMBL active molecules confirms their ability to act in the high-to-mid-micromolar concentration range (Table 6). The LE efficiency values are high compared to the often-used threshold value of 0.3 kcal·mol^−1^·atom^−1^, which renders them as promising fragment-like molecules. However, two caveats should be taken into consideration. Firstly, the values are based on the scoring function and docking position, not the experimental values. Secondly, most structures are known covalent SrtA inhibitors; thus, the direct use of those scaffolds for building novel noncovalent inhibitors is unlikely.

The size of the Ro3 subset of the ChEMBL molecules is small compared even to the compact site representation of the activated SrtA form. The molecules can generally adopt different configurations, forming binding modes with different site subpockets. Accordingly, the ligands in the docked binding modes fill all the pockets of the binding site in the best energy (Appendix A) and close binding modes. 

At the same time, the specific features of the binding site of SrtA seem to suggest a specific preferential composition and space configurations of the ligands. The site is a predominantly hydrophobic shallow pocket with practically a single polar residue available for directed polar interactions, Arg197. At physiological pHs, Arg197 is protonated and positively charged, thus providing the only option for ligands to form favorable directed interactions via salt bridging or as a hydrogen bond acceptor. However, Arg197 was shown previously to be more important as a hydrogen bond donor than as a charged species [56]. Based on the above SrtA binding site characterizations, an optimal ligand should be comprised of generally nonpolar (hydrophobic) fragments with few hydrogen bond acceptors. Additionally, the flat bottom of the site suggests that aromatic units might well fit the site pockets provided the scaffold of a ligand is able to properly position all substituents in the site.

A superposition of the most favorable binding modes of each Ro3-compliant StrA active ligands from ChEMBL generally confirms the assumption (Figure 9). Here, the carbonyl (sulfuryl) oxygens are highlighted as being the stronger hydrogen bond acceptors than the heteroatoms present in ligands in different chemical environments. The oxygen HB acceptors tend to occupy positions close to Arg197. Despite the hydrogen bonds with conventional geometries not being formed according to docking in most cases, two important factors should be considered. Firstly, purely electrostatic favorable interactions between the generally positive guanidine unit and Lewis donor oxygen atoms are formed. Secondly, the guanidine moiety of Arg197 is flexible within the environment of the site; thus, it can form conventional hydrogen bonds with the ligand in its different side chain conformations.

Remarkably, the results corroborating the hypothesis of a ligand should look like an “aprotic polar solvent” put forward based on the descriptor correlation and QSAR analysis above. Thus, the identified patterns based on the experimental data for ligand activity received additional confirmation in terms of the structure and intermolecular interactions. Therefore, this hypothesis could be used as a simple pharmacophore model to aid in designing new potent small molecule inhibitors.

#### 3.2.2. Prospective Ligand List

The docking energies for the most favorable modes found by docking for the prospective ligand list are provided in Table 7. The same coloring scheme as for the analysis of the Ro3 compatible ligands (Table 6) is used. The results will be analyzed in terms of the binding energy, LE and the overall reasonability of the docked ligand–receptor complex geometries. It should be stressed that, for the docking study, both possible enantiomer geometries were generated to investigate a possible enantiomer preference.

##### Energy-Based Ranking

For the majority of the structures tested, the predicted binding energy lies in the range corresponding to a mid-to-low micromolar range, with one structure, KUD649(7S), reaching a high nanomolar concentration (600 nM). That rendered many of those structures suitable for experimental screening for in vitro activity.

To select several structures with the maximum predicted binding energy for further analysis, an energy threshold of less than −7.32 kcal/mol (Kd estimate lower than 5 μM) was employed, resulting in the selection of KUD649 (both forms), KUD1135 (2S), KUD1044 (5R), KUD1133 (2S), KUD1036 (both forms), KUD1066, KUD833 (both forms), KUD1050 (7R), KUD834 (2R), KUD1022 (5R) and KUD759 (2S). This subset of structures tended to contain the largest molecules in the set, with a few notable exceptions. Firstly, the two largest molecules, KUD990 and KUD1008, appeared to be too large to fit into the compact binding site of the activated form of StrA, and despite their large sizes, did not display energies below the defined threshold. This corresponds well with the prediction that too large and too lipophilic ligands are unlikely to be active according to both the classification and regression models built. Secondly, both enantiomers of KUD649 displayed superior binding energy despite the sizes of the molecule being below the average size in the prospective molecule set, thus showing a decent LE value above the commonly used threshold of 0.3 kcal·mol^−1^·atom^−1^. It could be interpreted as roughly the optimal size of ca 27 heavy atoms appropriate for the binding site of the activated form of *S. aureus* SrtA is reached for KUD649. Interestingly, KUD649 was predicted as active by the classification models (with high rank values) and moderately active by the regression models, with a LASSO model prediction below average. The latter most probably reflects that KUD649 has a low structural similarity with the molecules of the training set taken from ChEMBL and/or a significantly nonlinear dependence of activity with respect to the descriptor values.

The worst binders according to the docking predictions are KUD1008, KUD1130, KUD529 (5R), KUD224 (both enantiomers), KUD225 (both enantiomers), KUD530 (5R), KUD138 (both enantiomers), KUD165 (both enantiomers) and KUD990. Most of these molecules are just small and have good specific activity, as judged by the LE values (below). KUD1008 is consistently predicted as not active by all the methods in the study. Evidently, it is too large for the activated form of the binding site of SrtA. The same is applicable to KUD990. Both KUD529 and KUD530, which are also larger on average than the other molecules in this group, were predicted to be active by the classification models (with almost the largest rank possible for being classified as still active) and intermediately active by the regression models.

##### Ligand Efficiency-Based Ranking

The smallest molecules (with the lowest number of heavy atoms) with significant specific activity towards *S. aureus* SrtA, which are the prospective starting points for hit-to-lead expansion and lead optimization, are the main objective of this study. To this end, the structures with high ligand efficiency (LE) values were analyzed. The most specifically active molecules with LE values greater than the threshold of 0.3 kcal·mol^−1^·atom^−1^ were both enantiomers of KUD718, KUD138, KUD165, KUD649, KUD1134, KUD224, KUD233, KUD1130 (single enantiomer) and KUD225. Despite the small sizes of these molecules, the predicted activity was in the mid-to-low micromolar range (Table 7). The relatively small sizes of the molecules presumably explained the negligible difference of the predicted binding energy for the pairs of the available enantiomers.

The worst molecules according to the LE metrics were KUD1008, KUD990, KUD1050 (both enantiomers), KUD529 (5R), KUD530 (5R), KUD759 (both enantiomers), KUD833 (both enantiomers) and KUD834 (both enantiomers). These were the largest molecules in the set, with the number of heavy atoms (NH) above 32, i.e., higher than a roughly detected optimal value of ca 27 for KUD649 presumably corresponding to the binding site maximum capacity, provided the form also fit well. Both the classification and regression models were also consistent with the docking results regarding these structures. All structures predicted as inactive were on this list. The low activity and, especially, the predicted ligand efficiency were shown for the regression model predictions.

##### Binding Modes Analysis

A binding mode analysis was made for the structures with the lowest sizes and highest LE (LE ≥ 0.29 kcal·mol^−1^·atom^−1^), since they are the most promising structure for further development, because growing additional substituents is generally easier than guessing how and which of the existing should be removed.

Due to the relatively small sizes, the ligands with the highest LE from the prospective ligand list form complexes with different spatial arrangements and close scoring function estimates. Several selected binding modes (as revealed by AutoDock Vina) of the intermediate size structures KUD1134 (2R), KUD718 (7S), KUD649 (7R) and KUD649 (7S), which almost fully occupy the binding site surface, are presented at Figure 10. In these modes, not only the ligand–receptor surface contact is reasonable, but also, the postulated above hypothesis regarding hydrogen bonding between Arg197 and ligand hydrogen acceptor oxygens seems to be confirmed at the structural level.

### 3.3. Selection of the Most Promising Ligands

In general, the choice of the most relevant ligands for further development in the absence of experimental data is a tricky task, especially if several decision metrics are involved. All the metrics influencing the decision are listed in Table 8 for all the structures from the prospective ligand list.

The most decisive metrics—and the most detailed ones in this study—are the predicted energy and LE from docking. Then, classification model’s predictions are taken into account, followed by the regression model predictions and their corresponding LE estimations. Finally, the drug-likeliness metrics—using the rule of five (Ro5) [27,28] criteria—are additionally considered to filter out structures that are less likely to be good starting points for further development. Special attention was paid to the small ligand sizes. Therefore, the LE metrics from the docking predictions were considered paramount.

Strikingly, setting the threshold value for LE ≥ 0.29 kcal·mol^−1^·atom^−1^ effectively selects the most promising structures fulfilling most of the criteria described (molecules in green in Table 8). The molecules left outside are mostly larger and more lipophilic than required for the site and correspond less to the *S. aureus* SrtA active structure from ChEMBL, as judged by both the classification and regression models. Several structures were filtered despite being predicted as intermediate strength binders due to their excessive sizes and complexities, which would make further developments harder. Interestingly, the number of heavy atoms, NH, appeared to be a good descriptor. The structures with NH > 27 (the value for KUD649 discussed earlier) were left outside, with a single exception of KUD1132 (NH = 20) considered as the moderately efficient ligand. 

Eventually, nine (Figure 11) out of 24 structures were selected for further research, which will include their synthesis and experimental testing against *S. aureus* SrtA activity. At this stage, both enantiomers will be considered; however, further optimization involving molecular modeling predictions will most likely result in a more emphasized preference of one of the enantiomers.

## 4. Conclusions

Within this work, several QSAR models—both classification and regression—were built based on the experimental data on known *S. aureus* SrtA inhibitors from ChEMBL. These models should be better considered as useful physicochemical filters rather than the precise and accurate prediction models due to the scarcity of the data they built upon and the inevitable bias in the chemical space of the known actives. In this work, the intentional crudeness of the models was balanced with reasonable accuracy by technical means of using the regularized versions of the algorithms and applying moderate values of regularization during the model learning.

Based on the analysis of the descriptor correlations and the coefficients at the descriptors in the final QSAR models, several generally useful SrtA activity patterns were identified. Firstly, excessive ligand size and lipophilicity adversely affect the likelihood of being active. The same is applied to the excessive number of chiral centers in a molecule. Secondly, charged and highly polar species are also not welcomed; however, a certain portion of the polar surface area should be present in a molecule to be active. Interestingly, a clear preference is made in favor of hydrogen bond acceptors, whereas the presence of hydrogen bond donors adversely affects the predicted activity. Thirdly, the combination of descriptors revealed was useful, and the literature data led us to a hypothesis that a proper fragment-sized ligand active against *S. aureus* SrtA should be similar to a “polar aprotic solvent”. It should possess a hydrophobic part complemented with a polar part represented with hydrogen bond acceptors, not donors, which could establish hydrogen bonds with Arg197. This hypothesis was generally confirmed later by the analysis of the docking results of the rule of three-compliant subset of SrtA active molecules from ChEMBL.

The obtained QSAR models were further used to select the most promising molecules from the list of ligands containing the dedicated molecular motif of 5-arylpyrrolidine-2-carboxylate or its derivatives. Additionally, molecular docking was conducted separately for each feasible enantiomer of all the molecules on the list. Since the binding site of the activated form of *S. aureus* SrtA (PDB:2KID) is compact, the molecules of relatively small sizes could produce reasonable binding modes and energies during the docking. For many of those ligands, the first or closely related in predicted binding energy modes revealed the above-mentioned binding mode, in which HB acceptor atoms were located in direct vicinity to the key Arg197 residue.

The most relevant nine structures out of 24 on the prospective ligand list were chosen based on their most fragment likeliness and consensus scoring using classification and regression models, as well as molecular docking studies. The most decisive criterion was the AutoDock Vina predicted ligand efficiency (LE) greater than or equal to 0.29 kcal·mol^−1^·atom^−1^, which brought the other criteria into consensus agreement. 

Despite both feasible enantiomers for each of the selected ligands generally adopting distinct binding modes according to docking, a significant energetic preference was not revealed in this study. This was a consequence of the relatively small sizes of the molecules. However, further rational optimization of each enantiomer should result in a more pronounced enantiomer preference.

The selected ligands will be synthesized and tested against *S. aureus* SrtA inhibitory activity with the aim of evaluating their prospects as starting points for promising leads for new noncovalently binding anti-infective drugs acting via the antivirulent mechanism.

We also believe that the patterns and dependencies put forward in this work could generally be useful in the field of developing novel *S. aureus* SrtA inhibitors.

## Figures and Tables

**Figure 1 molecules-26-07677-f001:**
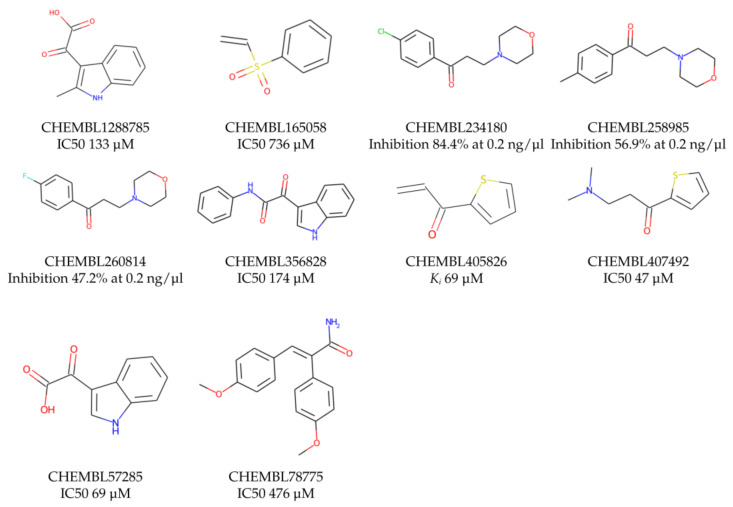
The Ro3-compliant fragment-sized known active compounds, along with their ChEMBL compound IDs and the experimental activities against *S. aureus* SrtA.

**Figure 2 molecules-26-07677-f002:**
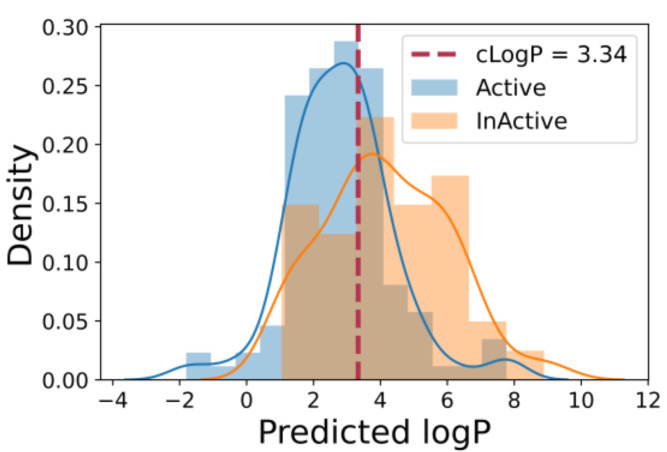
Distribution of lipophilicity among the active and inactive SrtA molecules.

**Figure 3 molecules-26-07677-f003:**
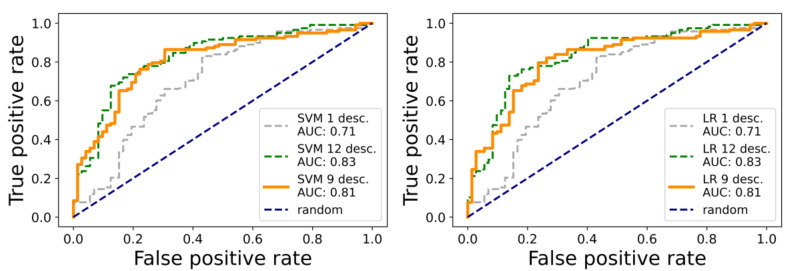
ROC curves for three types of SVM (**left**) and LR (**right**) models with different L1 regularization and effective descriptor numbers used.

**Figure 4 molecules-26-07677-f004:**
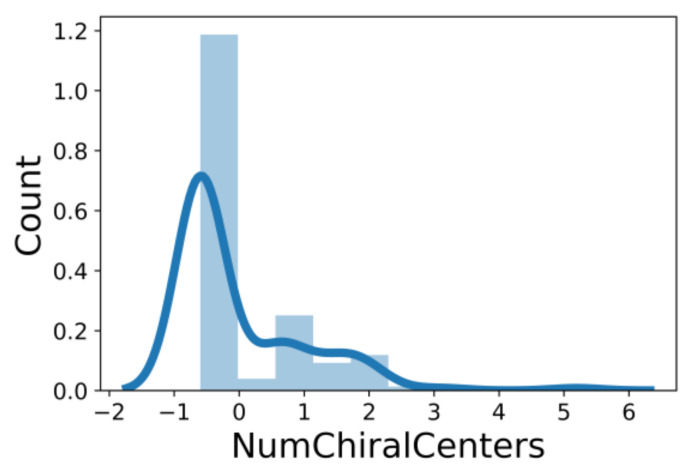
Normalized (zero mean and unit standard deviation) distribution of the number of chiral centers in the initial molecule set. Light blue bars are the data distribution, and the dark blue line is the gaussian kernel density estimate.

**Figure 5 molecules-26-07677-f005:**
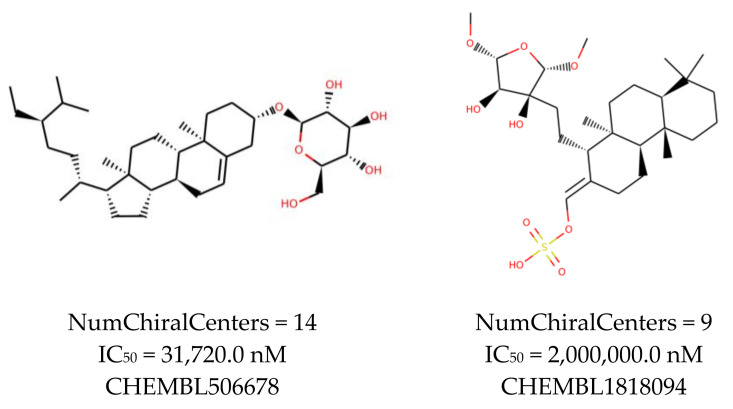
The structures excluded from the training set as having a nontypically high number of chiral centers per molecule.

**Figure 6 molecules-26-07677-f006:**
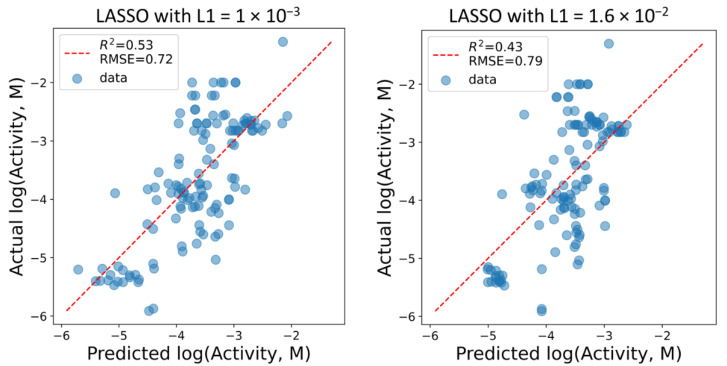
Performance of the LASSO regression models with slight (left) and intermediate (right) values of regularization parameter L1, resulting in 12- and 9-descriptor models.

**Figure 7 molecules-26-07677-f007:**
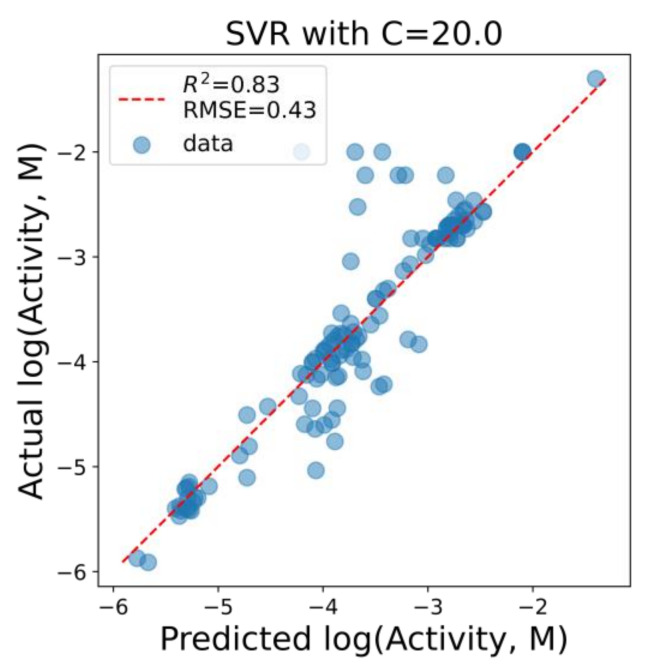
The performance of the SVR model on the training set of molecules.

**Figure 8 molecules-26-07677-f008:**
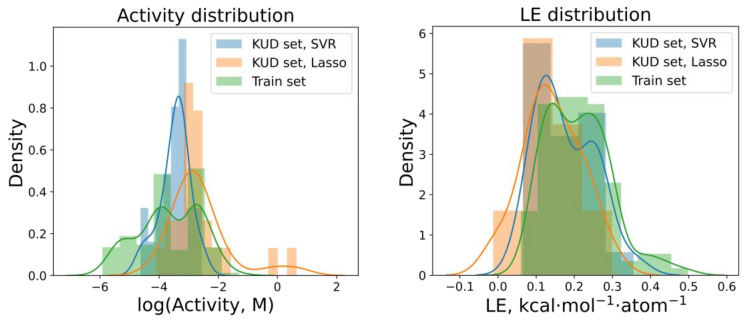
Comparison of the distributions of the activity and ligand efficiency (LE) both in the training set and predicted for the prospective ligand list (KUD set).

**Figure 9 molecules-26-07677-f009:**
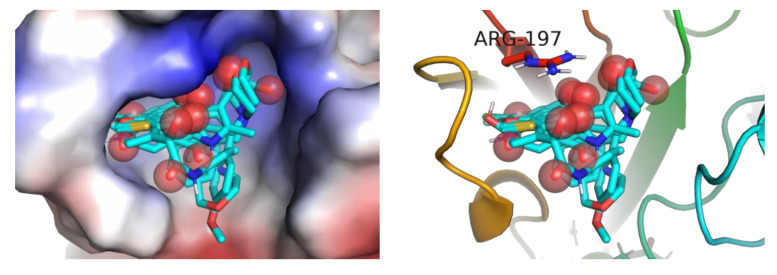
Superposition of the first binding modes of ligands from the Ro3 subset of active ligands in the binding site of *S. aureus* SrtA (PDB:2KID). Red spheres highlight the positions of carbonyl (sulfuryl) oxygens, which tend to be close to Arg197.

**Figure 10 molecules-26-07677-f010:**
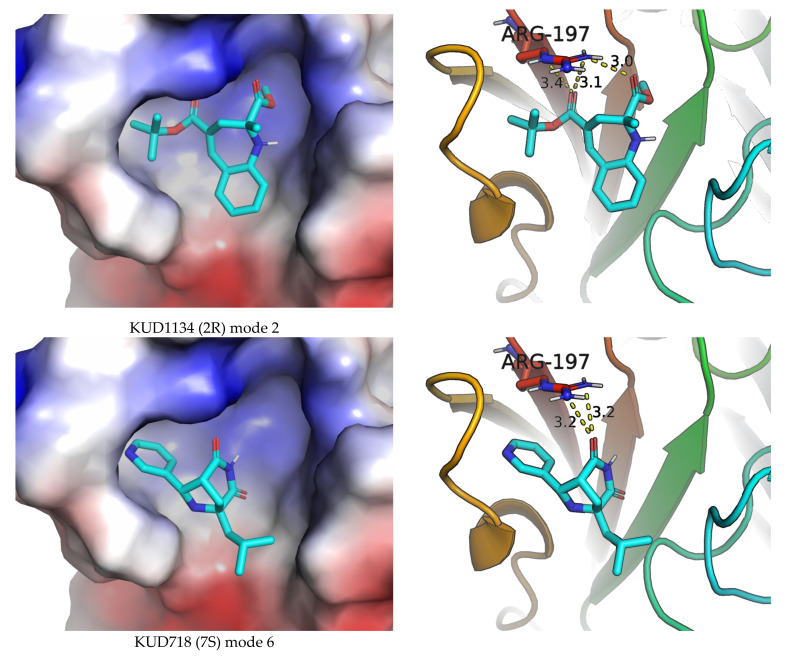
The selected binding modes for KUD1134 (2R), KUD718 (7S), KUD649 (7R) and KUD649 (7S) with potential hydrogen bonding interactions with Arg-197 of *S. aureus* SrtA. Left column—electrostatic surface site representation; right column—secondary structure site representation. All modes and structures are represented in the same view. The potential hydrogen bonds (between heteroatoms) are shown in yellow dash lines and Å distance.

**Figure 11 molecules-26-07677-f011:**
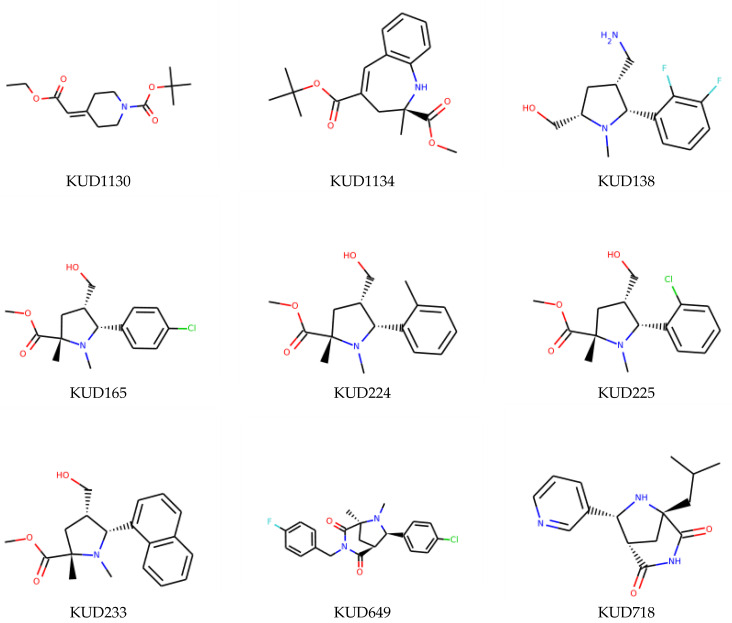
Ligands finally selected from the prospective ligand list based on consensus scoring. Random enantiomers are listed, since, at this stage, both possible enantiomers should be considered in further developments.

**Table 1 molecules-26-07677-t001:** The prospective list of synthetically feasible ligands with a dedicated molecular motif, 5-arylpyrrolidine-2-carboxylate, or closely related structures.

Num	Code	Enantiomer 1	Enantiomer 2
1	KUD1008	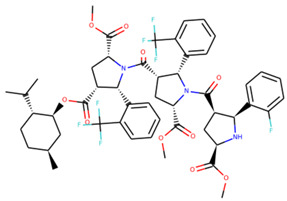	
2	KUD1022	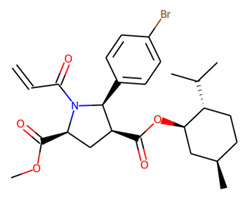 5R	
3	KUD1036	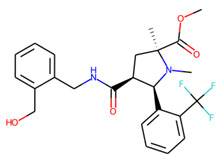 5R	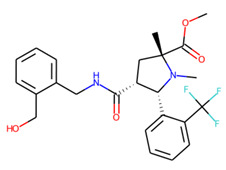 5S
4	KUD1044	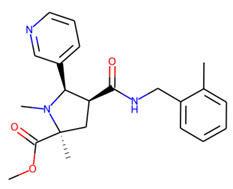 5R	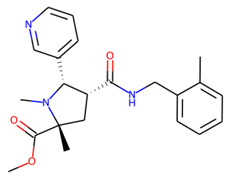 5S
5	KUD1050	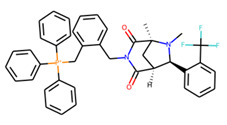 7R	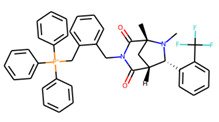 7S
6	KUD1066	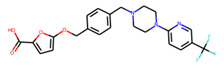	
7	KUD1130	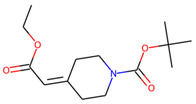	
8	KUD1132	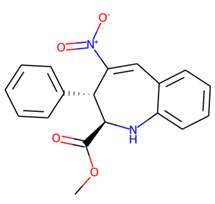 2R	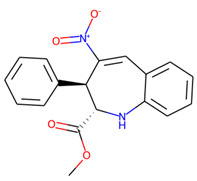 2S
9	KUD1133		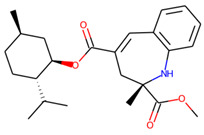 2S
10	KUD1134	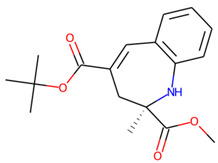 2R	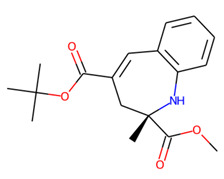 2S
11	KUD1135		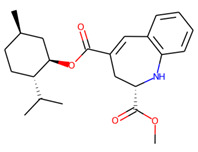 2S
12	KUD138	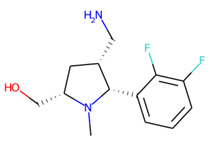 5R	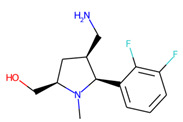 5S
13	KUD165	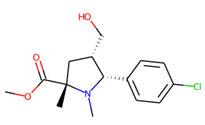 5R	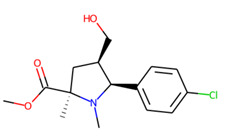 5S
14	KUD224	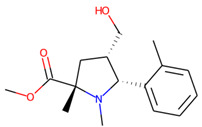 5R	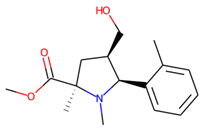 5S
15	KUD225	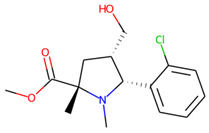 5R	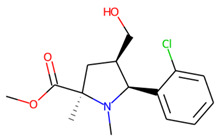 5S
16	KUD233	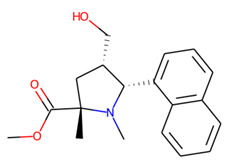 5R	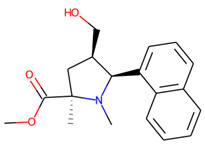 5S
17	KUD529	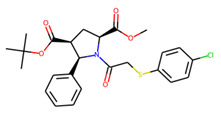 5R	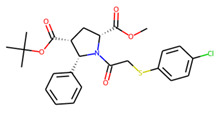 5S
18	KUD530	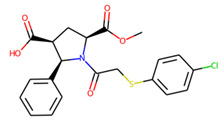 5R	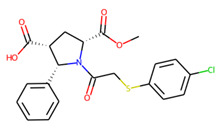 5S
19	KUD649	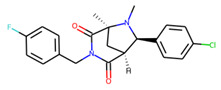 7R	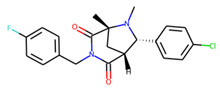 7S
20	KUD718	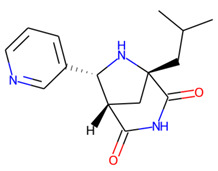 7R	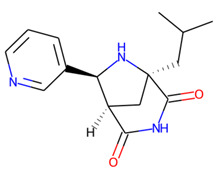 7S
21	KUD759	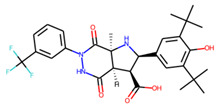 2R	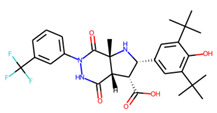 2S
22	KUD833	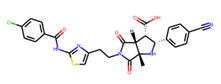 2R	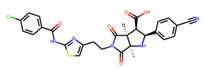 2S
23	KUD834	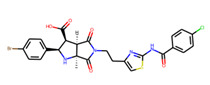 2R	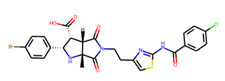 2S
24	KUD990	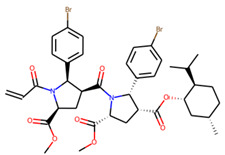	

**Table 2 molecules-26-07677-t002:** A set of descriptors used for building both classification and regression models.

Num	Descriptor	Meaning
1	MolLogP	cLogP
2	TPSA	Topological polar surface area
3	NHOHCount	Number of N and O hydrogen donors
4	NOCount	Number of N and O hydrogen acceptors
5	NumHAcceptors	Number of hydrogen acceptors
6	NumHDonors	Number of hydrogen donors
7	NumRotatableBonds	Number of rotatable bonds
8	NumHeteroatoms	Number of heteroatoms
9	FractionCSP3	Fraction of sp3 carbon atoms
10	BalabanJ	Topological connectivity index by Balaban
11	Chi0	Topological connectivity index
12	Chi1	Topological connectivity index
13	Chi3v	Topological connectivity index
14	Chi4v	Topological connectivity index
15	HallKierAlpha	Topological connectivity index by Kier-Hall
16	Kappa1	Topological connectivity index
17	Kappa2	Topological connectivity index
18	Kappa3	Topological connectivity index
19	BertzCT	Topological connectivity index
20	NumAliphaticRings	Number of aliphatic rings
21	NumAromaticRings	Number of aromatic rings
22	NumAtoms	Number of atoms
23	MolWt	Molecular weight
24	MolMR	Calculated molecular refractivity
25	MaxPartialCharge	Maximum partial charge (Gasteiger)
26	MinPartialCharge	Minimum partial charge (Gasteiger)
27	MaxAbsPartialCharge	Maximum modulo partial charge (Gasteiger)
28	NumChiralCenters	Number of chiral atomic centers

**Table 3 molecules-26-07677-t003:** Correlation coefficients between the initial descriptors and the log10(IC_50_).

#	Descriptor	Correlation Coefficient
1	MolLogP	0.235
2	NumHAcceptors	−0.011
3	NumHDonors	−0.069
4	NumRotatableBonds	0.103
5	NumHeteroatoms	0.032
6	FractionCSP3	−0.113
7	NumAliphaticRings	−0.215
8	NumAromaticRings	0.099
9	MaxPartialCharge	0.178
10	MinPartialCharge	−0.301
11	MaxAbsPartialCharge	0.329
12	NumChiralCenters	0.226

**Table 4 molecules-26-07677-t004:** The 12 most significant descriptors chosen for the quantitative regression models.

#	Descriptor	Feature Rank	Meaning
1	NumChiralCenters	1	Chirality
2	FractionCSP3	2	Chirality
3	MolWt	3	Size and shape
4	Kappa1	4	Size and shape
5	BertzCT	5	Size and shape
6	NOCount	6	Polarity
7	TPSA	7	Polarity
8	NumAliphaticRings	8	Size and shape
9	Chi3v	9	Size and shape
10	MolLogP	10	Size and shape
11	NumHeteroatoms	11	Polarity
12	NumHAcceptors	12	Polarity

**Table 5 molecules-26-07677-t005:** The coefficients of the 9-descriptor LASSO regression model.

#	Descriptor	Value (Lower; Upper Bounds) ^a^
1	MolLogP	0.00
2	TPSA	−0.50 (−0.95; −0.03)
3	NOCount	0.49 (0.00; 0.98)
4	NumHAcceptors	−0.14 (−0.47; 0.00)
5	NumHeteroatoms	0.00
6	FractionCSP3	−0.27 (−0.60; 0.00)
7	Chi3v	0.0013 (0.00; 0.49)
8	Kappa1	0.00
9	BertzCT	−0.14 (−0.45; 0.00)
10	NumAliphaticRings	−0.52 (−0.95; −0.30)
11	MolWt	0.32 (0.00; 0.86)
12	NumChiralCenters	0.69 (0.49; 0.86)

^a^ Confidence intervals at the 95% level obtained using the bootstrap method.

**Table 6 molecules-26-07677-t006:** The docking (AutoDock Vina) binding energies for the first binding modes of the Ro3 subset of the active ligands.

Num	ChEMBL Lig ID	Vina Energy, kcal/mol ^a^	NH	LE ^b^	Kd Predicted, μM
1	CHEMBL1288785	−6.3	15	0.42	27.5
2	CHEMBL165058	−5.3	11	0.48	145.8
3	CHEMBL234180	−6.3	17	0.37	27.5
4	CHEMBL258985	−6.3	17	0.37	27.5
5	CHEMBL260814	−6.5	17	0.38	19.7
6	CHEMBL356828	−6.6	20	0.33	16.7
7	CHEMBL405826	−4.5	9	0.50	553.1
8	CHEMBL407492	−4.9	12	0.41	284.0
9	CHEMBL57285	−6.3	14	0.45	27.5
10	CHEMBL78775	−6.7	21	0.32	14.1

^a^ The color scheme is −9.0—green, −6.5 (Kd ~ 20 μM)—white, −4.0—red; ^b^ in kcal·mol^−1^·atom^−1^, the threshold value of 0.25 is used as white for coloring, whereas 0.0 is used for red and 0.5 as green color thresholds.

**Table 7 molecules-26-07677-t007:** The best docking (AutoDock Vina) energies for the molecules from the prospective ligand list.

Num	Ligand	Vina Energy, kcal/mol ^a^	LE ^b^	NH	Kd Pred., μM
1	KUD1008	−4.1	0.06	71	1077.3
2	KUD1022_5r	−7.4	0.22	33	4.4
3	KUD1036_5r	−7.8	0.24	33	2.3
4	KUD1036_5s	−7.7	0.23	33	2.7
5	KUD1044_5r	−7.8	0.28	28	2.3
6	KUD1044_5s	−7.2	0.26	28	6.1
7	KUD1050_7r	−7.8	0.16	49	2.3
8	KUD1050_7s	−7.3	0.15	49	5.2
9	KUD1066	−7.8	0.24	33	2.3
10	KUD1130	−5.6	0.30	19	88.4
11	KUD1132_2r	−6.6	0.27	24	16.7
12	KUD1132_2s	−6.9	0.29	24	10.1
13	KUD1133_2s	−7.8	0.27	29	2.3
14	KUD1134_2r	−7.1	0.31	23	7.3
15	KUD1134_2s	−6.9	0.30	23	11.0
16	KUD1135_2s	−7.9	0.28	28	1.9
17	KUD138_5r	−6.1	0.34	18	38.4
18	KUD138_5s	−6.2	0.34	18	32.5
19	KUD165_5r	−6.4	0.32	20	23.3
20	KUD165_5s	−6.3	0.32	20	27.5
21	KUD224_5r	−6.1	0.30	20	38.4
22	KUD224_5s	−6.2	0.31	20	32.5
23	KUD225_5r	−6.1	0.30	20	38.4
24	KUD225_5s	−6.1	0.30	20	38.4
25	KUD233_5r	−6.9	0.30	23	10.1
26	KUD233_5s	−7.2	0.31	23	6.1
27	KUD529_5r	−6.2	0.19	33	32.5
28	KUD529_5s	−6.9	0.21	33	10.1
29	KUD530_5r	−6.6	0.20	29	16.7
30	KUD530_5s	−6.9	0.24	29	10.1
31	KUD649_7r	−8.3	0.31	27	1.0
32	KUD649_7s	−8.6	0.32	27	0.6
33	KUD718_7r	−7.0	0.35	20	8.6
34	KUD718_7s	−7.0	0.35	20	8.6
35	KUD759_2r	−6.9	0.17	40	10.1
36	KUD759_2s	−7.4	0.18	40	4.4
37	KUD833_2r	−7.8	0.20	39	2.3
38	KUD833_2s	−7.7	0.20	39	2.7
39	KUD834_2r	−7.5	0.20	38	3.7
40	KUD834_2s	−7.1	0.19	38	7.3
41	KUD990	−6.6	0.13	51	16.7

^a^ The color scheme is −9.0—green, −6.5 (Kd ~ 20 μM)—white, −4.0—red; ^b^ in kcal·mol^−1^·atom^−1^, the threshold value of 0.25 is used as white for coloring, whereas 0.0 is used for red and 0.5 as green color thresholds.

**Table 8 molecules-26-07677-t008:** The prospective ligand list, along with all the predictions made in the work for the consensus choices of the most promising ligands for further study.

								Docking			Classification		Regression ^c^		Regression LE ^d^	
#	Molecule	Ro5	MolLogP	HBA	HBD	MW	NH	Vina Energy, kcal/mol ^a^	LE ^b^	Kd Pred., μM	SVM, act.	LR, act.	Lasso, Log (Act., M)	SVR, Log (Act., M)	Lasso, LE	SVR, LE
1	KUD1008	FALSE	8.32	11	1	1004.0	71	−4.1	0.06	1077.3	0	1	0.6	−3.4	−0.01	0.07
2	KUD1022	FALSE	5.07	5	0	520.5	33	−7.4	0.22	4.4	0	0	−1.8	−2.6	0.07	0.11
3	KUD1036	TRUE	3.44	5	2	464.5	33	−7.8	0.24	2.3	1	1	−2.6	−3.2	0.11	0.14
4	KUD1044	TRUE	2.63	5	1	381.5	28	−7.7	0.28	2.7	1	1	−2.7	−2.7	0.13	0.13
5	KUD1050	FALSE	7.52	3	0	677.7	49	−7.6	0.16	3.2	0	0	−2.0	−3.2	0.06	0.09
6	KUD1066	TRUE	4.29	7	1	461.4	33	−7.8	0.24	2.3	1	1	−3.4	−3.2	0.14	0.13
7	KUD1130	TRUE	2.51	4	0	269.3	19	−5.6	0.29	88.4	1	1	−4.1	−3.6	0.30	0.26
8	KUD1132	TRUE	3.06	5	1	324.3	24	−6.8	0.28	12.0	1	1	−3.0	−4.5	0.17	0.26
9	KUD1133	TRUE	4.82	5	1	399.5	29	−7.8	0.27	2.3	0	0	−2.9	−3.1	0.14	0.15
10	KUD1134	TRUE	3.16	5	1	317.4	23	−7.0	0.30	8.6	1	1	−3.7	−4.2	0.22	0.25
11	KUD1135	TRUE	4.43	5	1	385.5	28	−7.9	0.28	1.9	0	0	−3.0	−3.3	0.15	0.16
12	KUD138	TRUE	1.28	3	2	256.3	18	−6.2	0.34	32.5	1	1	−3.3	−4.6	0.26	0.36
13	KUD165	TRUE	2.26	4	1	297.8	20	−6.4	0.32	23.3	1	1	−3.0	−3.9	0.21	0.27
14	KUD224	TRUE	1.91	4	1	277.4	20	−6.2	0.31	32.5	1	1	−3.2	−3.6	0.22	0.25
15	KUD225	TRUE	2.26	4	1	297.8	20	−6.1	0.31	38.4	1	1	−3.0	−3.8	0.21	0.26
16	KUD233	TRUE	2.76	4	1	313.4	23	−7.1	0.31	7.3	1	1	−2.9	−3.3	0.18	0.20
17	KUD529	TRUE	4.91	6	0	490.0	33	−6.6	0.20	16.7	1	1	−2.5	−3.4	0.10	0.14
18	KUD530	TRUE	3.65	6	1	433.9	29	−6.8	0.23	12.0	1	1	−2.7	−3.4	0.12	0.15
19	KUD649	TRUE	3.80	3	0	386.9	27	−8.5	0.31	0.7	1	1	−2.6	−3.9	0.13	0.20
20	KUD718	TRUE	1.17	4	2	273.3	20	−7.0	0.35	8.6	1	1	−3.5	−3.6	0.24	0.25
21	KUD759	FALSE	4.80	6	4	561.6	40	−7.2	0.18	6.1	0	0	−2.7	−3.3	0.09	0.11
22	KUD833	FALSE	3.25	9	3	564.0	39	−7.8	0.20	2.3	1	1	−2.7	−3.0	0.10	0.11
23	KUD834	FALSE	4.14	8	3	617.9	38	−7.3	0.19	5.2	1	1	−2.3	−3.0	0.08	0.11
24	KUD990	FALSE	6.96	8	0	830.6	51	−6.6	0.13	16.7	0	0	−0.2	−3.4	0.01	0.09

^a^ The color scheme is −9.0—green, −6.5 (Kd ~ 20 μM)—white, −4.0—red; ^b^ in kcal·mol^−1^·atom^−1^, the threshold value of 0.25 is used as white for coloring, whereas 0.0 is used for red and 0.5 as green color thresholds; ^c^ the threshold value of −3.0 kcal·mol^−1^ was used as white; ^d^ the value of 50% for the two columns was used as white.

## Data Availability

The main data presented in this study is contained within the Appendix A to this article.

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
