# Peer review of "Selection of Promising Novel Fragment Sized S. aureus SrtA Noncovalent Inhibitors Based on QSAR and Docking Modeling Studies"

_molecules, 2021, doi:10.3390/molecules26247677_

Round 1

Reviewer 1 Report

In attention of the manuscript authors,

In the “molecules-1469376” manuscript, the author has made substantial research efforts to identify new fragment-sized starting points to design new non-covalent S. aureus SrtA inhibitors by making use of the dedicated molecular motif, 5-arylpyrrolidine-2-carboxylate. To reach their goal,  in silico scenario involving known SrtA inhibitors from the ChEMBL database to build classification and regression QSAR models and molecular docking into the “activated” model of SrtA (PDB:2KID), has been applied. As a final result, 9 out of 24 of the prospective ligand list were chosen as promising S. aureus SrtA inhibitors based on their most fragment likeliness and consensus scoring as well as molecular docking studies.

The manuscript, if reduced in size and implicitly reorganized, along with the significant outcomes provided could be a real gain for researchers interested in developing promising new S. aureus SrtA candidates.

In this context, the potential impact of the manuscript results in the research world, and with all due respect to the impressive author’s work, the manuscript may be considered for publication in Molecules journal, after a major revision will be applied.

Recommendation:

  1. The manuscript is too long, has too many details, and is difficult to follow. It is strongly recommended that you keep only the relevant information, otherwise, the text loses its importance and significance.
  2. A workflow scheme will be very helpful for the reader.
  3. Table 1, please indicate the references for the selected compounds In this way, all the references listed, [27,28,29,30,31,32,33,34,35], are justified
  4. Table 4 and the paragraph before it, I do not understand the specification of these three rules, Lipinski, Weber, and Ghose, if they are not appropriate for this case. In the referee's opinion, a short note about the “rule of three” (Ro3) as a more appropriate filter than the other 3 for fragment-sized lead-like compounds is sufficient. If the authors decide to keep the paragraph then Table 4 could be moved in the supplementary material.
  5. Validation of predictive power is a crucial aspect of quantitative structure-activity relationship modeling, and without this step, the QSAR's importance is lost. I suggest the additional calculation of some validation parameters, eg: (1) all Tropsha criteria (2) Q2F1, Q2F2, Q2F3, the mean absolute error for the prediction set, the concordance correlation coefficient for prediction set, etc.) in order to verify the goodness of fit, robustness and predictive ability of QSAR models.
  6. In general, in QSAR, the corresponding errors are passed to the values of the coefficients.
  7. The manuscript contains a large number of Figures and Tables, perhaps some of them could be grouped. e.g. both Tables 6 and 10 could be merged into one, keeping, of course, the corresponding explanations and legend. Please, try to reduce the number or move some of the figures and tables into supplementary material. For example, Table 13 contains relevant information and provides an overview of all the results.
  8. The authors indicated in the docking section that “According to the criterium of the maximum predicted binding energy, the most favorable structures with predicted binding energy less than -7.32 kcal/mol are….). This -7.32 kcal/mol is considered a criterion? If yes, why this value? To whom does this value correspond?
  9. Figure 14, the values of selected H-bonds presented range between 2.1 Å -3.0 Å (please specify the values also in Table S5 of supplementary material). The distance for an H-bond is generally considered to be from 2.7 to 3.3 Å, with 3.0 Å to 3.2 Å being the most common value for protein and water hydrogen bonds but at 3.2 accepted as H-bond. Therefore, an H-bond at 2.1-2.4 Å it's a bit forced.
  10. Supplementary material, add the unit for activity (e.g. Table S1, S2).

Author Response

Response to Reviewer 1 Comments

  1. The manuscript is too long, has too many details, and is difficult to follow. It is strongly recommended that you keep only the relevant information, otherwise, the text loses its importance and significance.

Response 1: In order to improve readability we moved several tables and figures into supporting information

  1. A workflow scheme will be very helpful for the reader.

Response 2: We created a workflow scheme and propose to use it as a table of content (TOC) image for our paper.

  1. Table 1, please indicate the references for the selected compounds In this way, all the references listed, [27,28,29,30,31,32,33,34,35], are justified

Response 3: The synthetical experience from the listed references was used to derive partially virtual list of structures which can be synthetised in case of promising in silico predictions.

  1. Table 4 and the paragraph before it, I do not understand the specification of these three rules, Lipinski, Weber, and Ghose, if they are not appropriate for this case. In the referee's opinion, a short note about the “rule of three” (Ro3) as a more appropriate filter than the other 3 for fragment-sized lead-like compounds is sufficient. If the authors decide to keep the paragraph then Table 4 could be moved in the supplementary material.

Response 4: 1. We aimed to show that our correlations are close in spirit (and sober intention) to the known "rules" listed. 2. The table was moved to supplementary material as suggested.

  1. Validation of predictive power is a crucial aspect of quantitative structure-activity relationship modeling, and without this step, the QSAR's importance is lost. I suggest the additional calculation of some validation parameters, eg: (1) all Tropsha criteria (2) Q2F1, Q2F2, Q2F3, the mean absolute error for the prediction set, the concordance correlation coefficient for prediction set, etc.) in order to verify the goodness of fit, robustness and predictive ability of QSAR models.

Response 5: 1. We intended to build the simplest models (rules) and used regularization in order to a priori minimize the effect of possible overfitting. 2. But we agree a reader should be able to judge if we were successful in that. 3. For the reasons 1 and 2 we provide the simplest yet useful estimation – leave-one-out estimations, different for classification and regression models. 4. Moreover, the scans for regularization parameter were also provided in the supplementary material so that an interested reader could judge about how sensible was the final choice of those parameters in the main text.

  1. In general, in QSAR, the corresponding errors are passed to the values of the coefficients.

Response 6: An additional effort was made to estimate the confidence limits for the obtained models using the bootstrap method.

  1. The manuscript contains a large number of Figures and Tables, perhaps some of them could be grouped. e.g. both Tables 6 and 10 could be merged into one, keeping, of course, the corresponding explanations and legend. Please, try to reduce the number or move some of the figures and tables into supplementary material. For example, Table 13 contains relevant information and provides an overview of all the results.

Response 7: We agree on that and moved several figures and tables into supplementary material, in particular Tables 6 and 10 as suggested.

  1. The authors indicated in the docking section that “According to the criterium of the maximum predicted binding energy, the most favorable structures with predicted binding energy less than -7.32 kcal/mol are….). This -7.32 kcal/mol is considered a criterion? If yes, why this value? To whom does this value correspond?

Response 8: Agree, this part of text clearer was made clearer.

  1. Figure 14, the values of selected H-bonds presented range between 2.1 Å -3.0 Å (please specify the values also in Table S5 of supplementary material). The distance for an H-bond is generally considered to be from 2.7 to 3.3 Å, with 3.0 Å to 3.2 Å being the most common value for protein and water hydrogen bonds but at 3.2 accepted as H-bond. Therefore, an H-bond at 2.1-2.4 Å it's a bit forced.

Response 9: Agree. In fact, it was H---Acc distance, not usually implied Don(H)---Acc distance, since we believe the former better represents electrostatic interactions involved for not always conventional geometries of potential hydrogen bonds. However, it was fixed for better readability, since for the vast majority of readers the latter distance (heteroatom-heteroatom) is implied by “hydrogen bond distance”.

  1. Supplementary material, add the unit for activity (e.g. Table S1, S2).

Response 10: Agree, inserted.

Reviewer 2 Report

In the article entitled “Selection of promising novel fragment sized S. aureus SrtA noncovalent inhibitors based on QSAR and docking modeling studies” the authors performed QSAR and docking analysis of known SrtA inhibitors and molecular fragments to obtain new lead fragments and compounds to be synthesized and explored as non-covalent SrtA inhibitors. The idea behind the research is good, as well as the methodology and discussion, however there are some issues that need to be corrected before accepting this manuscript:

  1. The authors state that the SrtA enzyme is very flexible, even the active site shows significant variations in its size and shape, which is one of the reasons only covalent inhibitors have been reported so far. On the other side, the authors performed docking studies only on one enzyme conformation, “the activated form”, as the authors call it. How do the authors know that this conformation is the most prevalent one to which the ligands bind, it could be that this conformation is the result of ligand binding? More importantly, if the enzyme, and its active site, are so flexible, it would be very advisable to perform molecular dynamics simulations to observe if the obtained ligand-enzyme complexes are even stable.
  2. The second significant issue relates to QSAR models; the authors based their models on IC50 values. A comparison of IC50 is useless if the experiments are not performed with the same enzyme concentrations – if the authors want to compare these results, the IC50 values should be transformed into Ki values. Here are some links that explain the problem of IC50

https://journals.plos.org/plosone/article?id=10.1371/journal.pone.0061007

https://www.sciencedirect.com/science/article/pii/B9780123860095000096

I do believe the conclusions of this manuscript are valid, however the method of obtaining them is flawed and it needs to be corrected.

These were the two most significant issues that need to be corrected, other, less significant, comments are:

  1. Correct the capitalization of aureus in the title and in reference 17.
  2. In table 2, the word “polar” is missing in the TPSA explanation
  3. Also in table 2, in descriptor number 22, should it be “NumAtoms” instead of “HumAtoms”? The same goes for “HumHeteroAtoms” on page 10, line 261.
  4. On page 7, in lines 155-160, the authors write about “uncorrelated” and “less significant” descriptors. What was the criteria to determine if two descriptors are correlated and if a descriptor is significant or not?
  5. In lines 195-196, the authors wrote: “A conservative estimate of numerical value of activity was made by multiplying by 10 the activity data extracted from ChEMBL.” This multiplication needs a better explanation.
  6. On page 26, line 676, what do the authors mean by “the lightest molecules”? Are these the molecules with the lowest mass? Please rephrase this.
  7. I highly suggest that a native person (or a person with high English skills) checks and corrects the manuscript because, in the present form, the manuscript is hard to read and follow and I believe that many authors’ insights/discussion and conclusions were simply lost in translation. This is a well-performed research and it would be a shame if it didn’t reach its highest potential due to a language barrier.

Author Response

Response to Reviewer 2 Comments

  1. The authors state that the SrtA enzyme is very flexible, even the active site shows significant variations in its size and shape, which is one of the reasons only covalent inhibitors have been reported so far. On the other side, the authors performed docking studies only on one enzyme conformation, “the activated form”, as the authors call it. How do the authors know that this conformation is the most prevalent one to which the ligands bind, it could be that this conformation is the result of ligand binding? More importantly, if the enzyme, and its active site, are so flexible, it would be very advisable to perform molecular dynamics simulations to observe if the obtained ligand-enzyme complexes are even stable.

Response 1: We agree it is the most crucial issue for the target, but we want to make some finite and useful effort at a time. The main reasons for our simplification are as follows. 1) the activated form is the most structured (b6/b7) one (the least disordered), 2) the activated form is better suited for small molecules, 3) the activated form has been successfully used by others, 4) it is clearly related to SrtA function.

MD in this case is complex (state-of-the-art computations were made by others). We didn't want to overcomplicate the manuscript full of other details. Yes, we plan to use MD for further more in detail studies for the chosen ligands.

  1. The second significant issue relates to QSAR models; the authors based their models on IC50 values. A comparison of IC50 is useless if the experiments are not performed with the same enzyme concentrations – if the authors want to compare these results, the IC50 values should be transformed into Ki values. Here are some links that explain the problem of IC50

https://journals.plos.org/plosone/article?id=10.1371/journal.pone.0061007

https://www.sciencedirect.com/science/article/pii/B9780123860095000096

Response 2: Thanks for the valuable references! We wanted to use values from ChEMBL as is, since by design we aim to build the crudest but still useful models. We still would not like to manually curate all the data for the above reason, however, we will definitely apply the corrections kindly proposed by the reviewer in future works, where broader data set is available to build more accurate models.

I do believe the conclusions of this manuscript are valid, however the method of obtaining them is flawed and it needs to be corrected.

These were the two most significant issues that need to be corrected, other, less significant, comments are:

  1. Correct the capitalization of aureus in the title and in reference 17.

Response 3: Agree, fixed.

4. In table 2, the word “polar” is missing in the TPSA explanation

Response 4: Agree, fixed.

5. Also in table 2, in descriptor number 22, should it be “NumAtoms” instead of “HumAtoms”? The same goes for “HumHeteroAtoms” on page 10, line 261.

Response 5: Agree, fixed.

6. On page 7, in lines 155-160, the authors write about “uncorrelated” and “less significant” descriptors. What was the criteria to determine if two descriptors are correlated and if a descriptor is significant or not?

Response 6: Agree. 1. Fixed correlation threshold description was added to the manuscript 2. A descriptor is considered significant if it stays in the model at appreciable values of regularization penalty applied.

7. In lines 195-196, the authors wrote: “A conservative estimate of numerical value of activity was made by multiplying by 10 the activity data extracted from ChEMBL.” This multiplication needs a better explanation.

Response 7: Agree, made this part clearer in the text.

8. On page 26, line 676, what do the authors mean by “the lightest molecules”? Are these the molecules with the lowest mass? Please rephrase this.

Response 8: Agree, fixed. The most ligand efficient molecules were implied.

9. I highly suggest that a native person (or a person with high English skills) checks and corrects the manuscript because, in the present form, the manuscript is hard to read and follow and I believe that many authors’ insights/discussion and conclusions were simply lost in translation. This is a well-performed research and it would be a shame if it didn’t reach its highest potential due to a language barrier.

Response 9: Thanks for your high assessment of our work. We tried to improve English in the manuscript.

Reviewer 3 Report

Shulga and Kudryavtsev describe the application of in silico methods for selection of promising novel inhibitors of sortase A from S. aureus.

Major revisions:

  1. The selection of the positive and negative training sets is unconvincing. For example, in the negative set are included compounds with IC50>100 nM, which means that the structures did not reveal any activity at concentration of 100 nM used for screening. However, it is possible for these compounds to have activity at concentrations higher than 100 nM, e.g. at 200, 500, 1000 nM, i.e. they can turn out to be false negatives. In the same time, in the positive set are included compounds with IC50=1 000 000 nM (comp. 2, 6, 18) or even higher that make them practically non-binders, i.e. false positives.
  2. A gold standard in the QSAR analysis is the application of internal cross-validation in groups and validation with external test groups. Neither of these techniques for model validation is applied in the study.

Author Response

Response to Reviewer 3 Comments

  1. The selection of the positive and negative training sets is unconvincing. For example, in the negative set are included compounds with IC50>100 nM, which means that the structures did not reveal any activity at concentration of 100 nM used for screening. However, it is possible for these compounds to have activity at concentrations higher than 100 nM, e.g. at 200, 500, 1000 nM, i.e. they can turn out to be false negatives. In the same time, in the positive set are included compounds with IC50=1 000 000 nM (comp. 2, 6, 18) or even higher that make them practically non-binders, i.e. false positives.

Response 1: Thanks, it was fixed. 1. We believe we confused the reviewer by Table S2 caption, which incorrectly stated that the molecules are active. They are inactive. The caption was fixed. 2. Also there are different units used in the lists of actives and inactives used for classification purposes. As proposed by one of the reviewers, we provide units along with each value taken from ChEMBL in order to avoid comparison of values with different units.

  1. A gold standard in the QSAR analysis is the application of internal cross-validation in groups and validation with external test groups. Neither of these techniques for model validation is applied in the study.

Response 2: 1. We intended to build the simplest models (rules) and used regularization in order to a priori minimize the effect of possible overfitting. 2. But we agree a reader should be able to judge if we were successful in that. 3. For the reasons 1 and 2 we provide the simplest yet useful estimation – leave-one-out estimations, different for classification and regression models. 4. Moreover, the scans for regularization parameter were also provided in the supplementary material so that an interested reader could judge about how sensible was the final choice of those parameters in the main text.

Round 2

Reviewer 1 Report

In attention of the manuscript authors,

The authors responded satisfactorily to all referee’s requirements and made all the changes addressed both the manuscript and supplementary material. The computational workflow, the software’s together with the results analysis, interpretation and presentation employed to achieve their goal (the design of new non- covalent S. aureus SrtA inhibitors by making use of the dedicated molecular motif, 5-arylpyrrolidine-2-carboxylate) recommend the present manuscript as having scientific soundness and great interest for the readers.

In this context, I agree that the manuscript should be accepted for publication in Molecules journal, in this uploaded version.

Author Response

Thank you.

Reviewer 2 Report

In the revised version of the article entitled “Selection of promising novel fragment sized S. aureus SrtA noncovalent inhibitors based on QSAR and docking modeling studies” the authors addressed all my minor concerns; however, the two big concerns (MD and QSAR) remain unanswered:

MD

Comment 1:

The authors state that the SrtA enzyme is very flexible, even the active site shows significant variations in its size and shape, which is one of the reasons only covalent inhibitors have been reported so far. On the other side, the authors performed docking studies only on one enzyme conformation, “the activated form”, as the authors call it. How do the authors know that this conformation is the most prevalent one to which the ligands bind, it could be that this conformation is the result of ligand binding? More importantly, if the enzyme, and its active site, are so flexible, it would be very advisable to perform molecular dynamics simulations to observe if the obtained ligand-enzyme complexes are even stable.

Response 1:

We agree it is the most crucial issue for the target, but we want to make some finite and useful effort at a time. The main reasons for our simplification are as follows. 1) the activated form is the most structured (b6/b7) one (the least disordered), 2) the activated form is better suited for small molecules, 3) the activated form has been successfully used by others, 4) it is clearly related to SrtA function.

MD in this case is complex (state-of-the-art computations were made by others). We didn't want to overcomplicate the manuscript full of other details. Yes, we plan to use MD for further more in detail studies for the chosen ligands.

Comment on response 1:

I do realize that performing MD simulations takes time and skill maybe not readily available to the authors, but be very careful when drawing conclusions from docking results, they can be very unreliable (as is nicely explained here https://doi.org/10.1016/j.tips.2014.12.001). Since the authors’ protein is very flexible, this becomes even more important. Additionally, the authors did not mention if they used flexible docking or not, so I am assuming they did not, which would help obtain more reliable docking data, especially with flexible protein, such as in this case.

QSAR

Comment 2:

The second significant issue relates to QSAR models; the authors based their models on IC50 values. A comparison of IC50 is useless if the experiments are not performed with the same enzyme concentrations – if the authors want to compare these results, the IC50 values should be transformed into Ki values. Here are some links that explain the problem of IC50

https://journals.plos.org/plosone/article?id=10.1371/journal.pone.0061007

https://www.sciencedirect.com/science/article/pii/B9780123860095000096

Response 2: Thanks for the valuable references! We wanted to use values from ChEMBL as is, since by design we aim to build the crudest but still useful models. We still would not like to manually curate all the data for the above reason, however, we will definitely apply the corrections kindly proposed by the reviewer in future works, where broader data set is available to build more accurate models.

Comment on response 2:

In the response the authors said: “We still would not like to manually curate all the data…” Transforming data from IC50 to Ki is not data curation, it is converting raw data into something useful. Using IC50 values from experiments which were not performed under the same conditions is a bad practice and is simply wrong. It does not just lead to “less accurate models”, as the authors imply, it leads to wrong models which should not be used, for any purpose. I realize it would take a lot of time to generate and explain results of new models, but results based on bad models are bad results.

Additional comment:

The authors added a few lines about leave-one-out validation. With this number of active (118 in classification, 86 in regression) and inactive molecules (72 in classification, 45 in regression), cross-validation should have been used instead. These numbers are too high to see a significant influence of one molecule to the resulting model.

As a conclusion, the authors have not done anything to address my concerns. The manuscript is of good quality and has much potential, but unfortunately, a combination of inflexible docking without even minimization of the obtained conformations to accommodate for the flexibility of the protein (which could be overlooked if it was the only issue) with QSAR results based on bad models still does not warrant acceptance of this article.

Author Response

Response to reviewer 2 comments

MD

Comment 1:

The authors state that the SrtA enzyme is very flexible, even the active site shows significant variations in its size and shape, which is one of the reasons only covalent inhibitors have been reported so far. On the other side, the authors performed docking studies only on one enzyme conformation, “the activated form”, as the authors call it. How do the authors know that this conformation is the most prevalent one to which the ligands bind, it could be that this conformation is the result of ligand binding? More importantly, if the enzyme, and its active site, are so flexible, it would be very advisable to perform molecular dynamics simulations to observe if the obtained ligand-enzyme complexes are even stable.

Response 1:

We agree it is the most crucial issue for the target, but we want to make some finite and useful effort at a time. The main reasons for our simplification are as follows. 1) the activated form is the most structured (b6/b7) one (the least disordered), 2) the activated form is better suited for small molecules, 3) the activated form has been successfully used by others, 4) it is clearly related to SrtA function.

MD in this case is complex (state-of-the-art computations were made by others). We didn't want to overcomplicate the manuscript full of other details. Yes, we plan to use MD for further more in detail studies for the chosen ligands.

Comment on response 1:

I do realize that performing MD simulations takes time and skill maybe not readily available to the authors, but be very careful when drawing conclusions from docking results, they can be very unreliable (as is nicely explained here https://doi.org/10.1016/j.tips.2014.12.001). Since the authors’ protein is very flexible, this becomes even more important. Additionally, the authors did not mention if they used flexible docking or not, so I am assuming they did not, which would help obtain more reliable docking data, especially with flexible protein, such as in this case.

Response to Comment on response 1: The question here is not in skills, which are available to the authors (15+ years in MD for commercial purposes), but rather in the intrinsic complexity of the problem under study. As previous in depth and state-of-the-art studies revealed [e.g. 10.1002/pro.2168, 10.1016/j.bpj.2015.08.039], the b7/b8 and especially b6/b7 loops, forming the walls of the binding site, undergo large scale motions at microsecond to millisecond timescale. What is worse, the disorder-to-order transition is taking place for the b6/b7 loop. The authors conclude that multiple favorable binding modes are available for the native substrate, which eventually accommodates to the catalytically relevant mode. Unfortunately, even those very laborious state-of-the-art MD simulations have not resulted in any significant and so much awaited breakthroughs in the field. We intentionally used (as pointed earlier) structure (PDB:2KID) with the most ordered in b6/b7 loop, which is also the most relevant for the other reasons provided in the manuscript. Most notably, it was successfully used to rationalize the experimental activity of a series of reversible inhibitors published in J. Med. Chem. [10.1021/acs.jmedchem.0c00803]. They are also close to fragment size. Thus, concerning MD, we support the Reviewer 2 opinion that generally it should be used whenever possible to validate the docking binding modes, but in this specific case, a superficial MD study would not be convincing, whereas a sound MD study of this system definitely requires a dedicated project with a separate paper as a result.

Concerning flexible docking, for the reasons described above we intentionally used rigid docking. The main theoretical reason is the described above flexibility could not be treated decently with any available docking technique/program. Most programs imply by flexible docking just the ability to sample several side chains of the target residues, which is definitely inappropriate to sample conformations spanning the conformational space available to disorder-to-order transitions of the b6/b7 loop. A few flexible docking programs/protocols treat the large scale motions of the backbone coordinates as quadratic fluctuations about the initial coordinates. This approach helps to more properly model e.g. the movements of seven alpha-helices of the GPCR receptors, however again it is still not sufficient to adequately model the described b6/b7 loop movement. What is left is to model the system with the state-of-the-art MD approaches since even very long simple runs currently available do not guarantee the necessary transitions are occurred and brought in equilibration, because of the micro- to milli-second characteristic timescales of the transitions involved.

The practical reason is that rigid docking with the compact “activated” enzyme form, closely represented by PDB:2KID (which we use), have been successfully used earlier (e.g. in above mentioned [10.1021/acs.jmedchem.0c00803] work).

Again, we agree with the concerns of the Reviewer 2 in general, but in this particular case we believe we made a judicious choice of the methods for the reasons described above.

QSAR

Comment 2:

The second significant issue relates to QSAR models; the authors based their models on IC50 values. A comparison of IC50 is useless if the experiments are not performed with the same enzyme concentrations – if the authors want to compare these results, the IC50 values should be transformed into Ki values. Here are some links that explain the problem of IC50

https://journals.plos.org/plosone/article?id=10.1371/journal.pone.0061007

https://www.sciencedirect.com/science/article/pii/B9780123860095000096

Response 2: Thanks for the valuable references! We wanted to use values from ChEMBL as is, since by design we aim to build the crudest but still useful models. We still would not like to manually curate all the data for the above reason, however, we will definitely apply the corrections kindly proposed by the reviewer in future works, where broader data set is available to build more accurate models.

Comment on response 2:

In the response the authors said: “We still would not like to manually curate all the data…” Transforming data from IC50 to Ki is not data curation, it is converting raw data into something useful. Using IC50 values from experiments which were not performed under the same conditions is a bad practice and is simply wrong. It does not just lead to “less accurate models”, as the authors imply, it leads to wrong models which should not be used, for any purpose. I realize it would take a lot of time to generate and explain results of new models, but results based on bad models are bad results.

Response to comment on response 2: There are two main arguments, why IC50 to Ki conversion should be done. First, it’s necessary if we use the mixed set of IC50 and Ki date to build the model. It’s not our case, we use IC50 for regression model building as the most abundant source of activity data available in ChEMBL. Second, the concentration of an enzyme is not sufficient enough to reveal in full the true (high) activity of a ligand. This is a good argument. But a decent part of the ligands act irreversibly, which violates the theoretical basis of this conversion. Still if they are active against SrtA they should bind near the active site. At least several papers show that the inhibitors studied are more active (in IC50 sense) than the non-selective p-Hydroxymercuribenzoic acid, irreversible inactivating the sulfhydryl groups of enzymes (such as Cys184 of SrtA).

For the rest of reversible inhibitors the Km/Kd data which enters the conversion model on input are not available for the structures which do have IC50 data in ChEMBL. So we are technically unable to recalculate the values. It’s reasonable from the point of view of the practical drug discovery, since the methods with higher throughput are used earlier to form a funnel to screen the structures, for which more laborious methods are applied if they have exhibited a satisfactory level of activity at previous stages. No surprise then, that in detail kinetic studies (giving e.g. Km values) are far more rarely occurred in ChEMBL (and in real life drug discovery process), than IC50 data.

If more reliable data for s. aureus SrtA inhibition were available, we would be the first to use them. However, in this case the data are too scarce, so we try to get some value out of them in order to rank our structures. The top ranked our structures will be more thoroughly studied then. We adhere here to the principle that “all models are wrong, but some are useful”. Moreover we think the way we process the data in these circumstances is decent.

Additional comment:

The authors added a few lines about leave-one-out validation. With this number of active (118 in classification, 86 in regression) and inactive molecules (72 in classification, 45 in regression), cross-validation should have been used instead. These numbers are too high to see a significant influence of one molecule to the resulting model.

Response to additional comment: Despite we believe the structures are too dissimilar to deserve general cross-validation, we tested 5-fold cross-validation for the production 9-descriptors regression LASSO model. We produced randoms splits 10 times since we expect that the composition of the train and test sets would sufficiently affect the results for such a molecule set. Leave-one-out estimates are: R2_LOO = 0.34, RMSE_LOO = 0.85, MAE_LOO = 0.68. 5-fold cross-validation estimates are: R2_cv5 = 0.27 (+/- 0.1), RMSE_cv5 = 0.85 (+/- 0.04), MAE_cv5 = 0.69 (+/- 0.04). Thus we see the qualitative and quantitative agreement, despite in the case of cross-validation the spread is expectedly higher. Thus exactly the same conclusions would be derived in the paper if the cross-validation were used.

Reviewer 3 Report

Shulga and Kudryavtsev considered my recommendations. The models have moderate predictability due to non-homogeneous initial data but could be used as a starting point for designing of novel molecules.

Author Response

Thank you.

Round 3

Reviewer 2 Report

I thank the authors for their detailed and well-argued response, especially for the MD/flexible docking comment. I do not completely agree with the argumentation for the IC50 comment. I also have to say that there are many articles/models published based on IC50 data, which are “more” wrong than these models, however this is not a good argument on itself. On the other hand, I do believe that the authors are knowledgeable about this field and have made their decisions consciously. Therefore, and as can be seen from my ratings of the manuscript and as I said to the editor, I am not against publication of this article, and I will not push this matter further. As a side note, I do think the problem of MD simulations for SrtA is an interesting and important one and the authors should, if they are willing, consider adding a few sentences about it in the article.

Author Response

Response to Reviewer 2 Comments and Suggestions for Authors

I thank the authors for their detailed and well-argued response, especially for the MD/flexible docking comment. I do not completely agree with the argumentation for the IC50 comment. I also have to say that there are many articles/models published based on IC50 data, which are “more” wrong than these models, however this is not a good argument on itself. On the other hand, I do believe that the authors are knowledgeable about this field and have made their decisions consciously. Therefore, and as can be seen from my ratings of the manuscript and as I said to the editor, I am not against publication of this article, and I will not push this matter further. As a side note, I do think the problem of MD simulations for SrtA is an interesting and important one and the authors should, if they are willing, consider adding a few sentences about it in the article.

Response to the Comments and suggestions:

Thank you. Yes, the choice was conscious due to lack of more reliable data. We believe we were able to extract something valuable even from these data.

We added a short notion of the underlying dynamic complexity of the LPXTG-SrtA system in Introduction.